

The Atmospheric Potential Oxygen forward Model Intercomparison Project
(APO-MIP1): Evaluating simulated atmospheric transport of air-sea gas exchange
tracers and APO flux products
Yuming Jin[1,2], Britton B. Stephens[1], Matthew C. Long[3], Naveen Chandra[4], Frédéric Chevallier[5],
Joram J.D. Hooghiem[6], Ingrid T. Luijkx[6], Shamil Maksyutov[7], Eric J. Morgan[8], Yosuke Niwa[7],
Prabir K. Patra[4,9], Christian Rödenbeck[10], Jesse Vance[11]
1. Earth Observing Laboratory, NSF National Center for Atmospheric Research, Boulder, CO 80301,
USA
2. Advanced Study Program, NSF National Center for Atmospheric Research, Boulder, CO 80301
3. [C]Worthy, Boulder, CO 80302, USA
4. Research Institute for Global Change, Japan Agency for Marine-Earth Science and Technology,
Yokohama, 236-0001, Japan
5. Laboratoire des Sciences du Climat et de l'Environnement, LSCE/IPSL, CEA-CNRS-UVSQ,
Université Paris-Saclay, Gif-sur-Yvette, F-91198, France
6. Wageningen University, Environmental Sciences Group, Wageningen, 6700AA, The Netherlands
7. Earth System Division, National Institute for Environmental Studies, Tsukuba, 305-8506, Japan
8. Geosciences Research Division, Scripps Institution of Oceanography, University of California, San
Diego, La Jolla, CA 92093, USA
9. Seto Inland Sea Carbon Neutral Research Center, Hiroshima University, Higashi-Hiroshima, 739-8529,
Japan
10. Max Planck Institute for Biogeochemistry, Jena, 07745, Germany
11. Ebb Carbon, San Carlos, CA 94070, USA
Correspondence: Yuming Jin (yumingjin@ucar.edu)



**Abstract**

Atmospheric Potential Oxygen (APO, defined as $O_2 + 1.1 \times CO_2$) is a tracer of air-sea $O_2$ exchange, exhibiting strong seasonal variability over mid-to-high latitudes. We present results from the first version of Atmospheric Potential Oxygen forward Model Intercomparison Project (APO-MIP1), which forward transports three air-sea APO flux products in eight atmospheric transport models or model variants, aiming to evaluate atmospheric transport and flux representations by comparing simulations against surface station, airborne, and shipboard observations of APO. We find significant spread and bias in APO simulations at eastern Pacific surface stations, indicating inconsistencies in representing vertical and coastal atmospheric mixing. A framework using airborne APO observations demonstrates that most atmospheric transport models (ATMs) participating in APO-MIP1 overestimate tracer diffusive mixing across moist isentropes (i.e., diabatic mixing) in mid-latitudes. This framework also enables us to isolate ATM-related biases in simulated APO distributions using independent mixing constraints derived from moist static energy budgets from reanalysis, thereby allowing us to assess large-scale features in air-sea APO flux products. Furthermore, shipboard observations show that ATMs are unable to reproduce seasonal APO gradients over Drake Passage and near Palmer Station, Antarctica, which could arise from uncertainties in APO fluxes or model transport. The transport simulations and flux products from APO-MIP1 provide valuable resources for developing new APO flux inversions and evaluating ocean biogeochemical processes.

**Short Summary**

We carry out a comprehensive atmospheric transport model (ATM) intercomparison project. This project aims to evaluate errors in ATMs and three air-sea $O_2$ exchange products by comparing model simulations with observations collected from surface stations, ships, and aircraft. We also present a model evaluation framework to independently quantify transport-related and flux-related biases that contribute to model-observation discrepancies in atmospheric tracer distributions.



## 1. Introduction

Atmospheric potential oxygen (APO), defined as the weighted sum of $O_2$ and $CO_2$ concentration ($APO = O_2 + 1.1\ CO_2$), is an important tracer of fossil fuel burning and ocean biogeochemical processes (Stephens et al., 1998). APO is intended to be unaffected by terrestrial photosynthesis and respiration due to the cancellation of $O_2$ and $CO_2$ exchange at an approximate $O_2$:C ratio of -1.1 (Severinghaus, 1995). APO exhibits a large seasonal cycle driven mainly by air-sea $O_2$ exchange due to upper ocean biological activities, deep water ventilation, and thermally induced $O_2$ solubility changes. Seasonal APO variability is also slightly affected by the air-sea exchange of $CO_2$ and $N_2$ (Manning & Keeling, 2006). APO is decreasing in the atmosphere due to fossil fuel combustion, which acts as an $O_2$ sink and $CO_2$ source with a more negative $O_2$:$CO_2$ ratio (global mean ~ -1.4) compared to the assumed -1.1 ratio from terrestrial processes. Although fossil fuel combustion contributes to an annual interhemispheric gradient that has lower APO in the Northern Hemisphere, it has only a minor effect on the seasonal cycle globally (Keeling & Manning, 2014).

APO measurements provide critical constraints on seasonal air-sea $O_2$ fluxes, which have been used to estimate air-sea gas exchange rates and ocean net community production (NCP), and to benchmark marine NCP in Earth system models (Keeling et al., 1998; Naegler et al., 2007; Nevison et al., 2012, 2015, 2016, 2018). APO has been used for improved partitioning of ocean and land carbon sinks (Friedlingstein et al., 2025; Manning & Keeling, 2006), to constrain ocean heat uptake and meridional heat transport (Resplandy et al., 2016, 2019), and to quantify fossil fuel emissions (Pickers et al., 2022; Rödenbeck et al., 2023). APO measurements are available at surface stations (Adcock et al., 2023; Battle et al., 2006; Goto et al., 2017; Keeling & Manning, 2014; Manning & Keeling, 2006; Nguyen et al., 2022; Tohjima et al., 2019), on ship transects (Ishidoya et al., 2016; Pickers et al., 2017; Stephens et al., 2003; Thompson et al., 2007; Tohjima et al., 2012, 2015, 2024), and from aircraft (Bent, 2014; Ishidoya et al., 2012; Jin et al., 2023; Langenfelds, 2002; Morgan et al., 2021; Stephens et al., 2018, 2021).

Global-scale air-sea APO fluxes have been estimated from APO measurements and an ATM within a Bayesian inversion framework (Rödenbeck et al., 2008). ATMs are also used to forward transport APO fluxes simulated from ocean biogeochemistry models (Carroll et al., 2020; Yeager et al., 2022) and surface ocean dissolved oxygen (DO) measurements (Garcia & Keeling, 2001;



Najjar & Keeling, 2000) to compare with atmospheric observations, providing a basis for model
and flux product evaluation (Jin et al., 2023; Keeling et al., 1998; Stephens et al., 1998).
However, using atmospheric data to evaluate flux products and to derive fluxes through
inversion is fundamentally limited by biases in ATMs, particularly in their representation of
vertical transport and diabatic mixing (Jin et al., 2024; Naegler et al., 2007; Nevison et al., 2008;
Schuh et al., 2019; Schuh & Jacobson, 2023; Stephens et al., 2007). The systematic uncertainties
in transport modeling limit inversions of APO, $CO_2$, and other greenhouse gases, underscoring
the need for independent transport bias assessments to advance global carbon budget constraints.
To address uncertainty in ATMs for studying large-scale tracer atmospheric transport and the
corresponding surface fluxes, several community model intercomparison (TransCom) projects
have been established for various tracers including $CO_2$ (Baker et al., 2006; Gurney et al., 2003,
2004; Law et al., 2008; Patra et al., 2008), $N_2O$ (Thompson et al., 2014), $SF_6$ (Denning et al.,
1999), $SF_6$ and $CH_4$ jointly (Patra et al., 2011), as well as an age of air tracer (Krol et al., 2018).
Blaine (2005) coordinated a TransCom $O_2$ experiment to compare model simulations of the $O_2$
seasonal cycle across the Scripps $O_2$ network. While this experiment provided valuable initial
insights into ATM performance in simulating atmospheric $O_2$ from ocean fluxes, substantial
advances in ATMs and more data collected also from aircraft and ships since then motivate an
updated intercomparison study with more extensive model-data comparisons and analyses. More
recently, $CO_2$ inversion intercomparisons have been coordinated through the OCO-2 MIP
(Crowell et al., 2019; Peiro et al., 2022; Byrne et al., 2023) and the Global Carbon Project (e.g.,
Friedlingstein et al., 2025). These experiments reveal substantial spread in forward tracer (e.g.,
$CO_2$) atmospheric distribution and inverted surface fluxes, driven by different ATMs and
inversion setups. The spread in forward transport simulations stems from multiple factors,
including the choice of wind fields from various reanalysis products or online simulation,
regridding fine resolution meteorological data to coarse model grids, the advection scheme that
governs large-scale mixing, and parameterized sub-grid processes, such as boundary layer
mixing and deep convection. Despite the complexity of different transport pathways, long-lived
tracers (e.g., $CO_2$ and $O_2$) at mid-latitudes tend to show tracer distributions that are aligned with
moist potential temperature ($\theta_e$) surfaces. This is because $\theta_e$ surfaces are preferential surfaces for
mixing, leading to rapid along-$\theta_e$ mixing and slow cross-$\theta_e$ mixing (Bailey et al., 2019; Jin et al.,
2021; Miyazaki et al., 2008; Parazoo et al., 2011).



It is a critical challenge to accurately quantify the rate-limiting cross-$\theta_e$ mixing time-scales, which are largely driven by diabatic processes including moist convection and radiative cooling. Here, we define "diabatic mixing rates" as diffusivities that are inversely related to cross-$\theta_e$ mixing time-scales. These mixing rates are important for determining the large-scale tracer distribution in ATMs. Jin et al. (2024) established a framework to calculate cross-$\theta_e$ mixing rates from ATMs and moist static energy (MSE) budgets from reanalysis based on a mass-indexed isentropic coordinate called $M_{\theta e}$ (Jin et al., 2021). This framework also allows cross-$\theta_e$ tracer gradients from airborne observations to provide independent constraints on diabatic mixing. Jin et al. (2024) tested four ATMs used in $CO_2$ inversions, showing that these models tend to have too fast mixing in the mid-latitudes of the Southern Hemisphere in the austral summer. The too fast mixing is also confirmed by the fact that models simulate smaller $CO_2$ gradients compared to airborne observations, which is an independent constraint on the mixing rate. The mixing rate constraint and $CO_2$ gradient constraint also have implications for biases in the inverse model estimates, indicating a too large summer-time Southern Ocean (SO) $CO_2$ sink. This framework provides a system for independently evaluating transport simulations and flux estimates.

Previous TransCom experiments focused primarily on tracers that only have significant sources and sinks over the land, and large seasonal flux cycles tied to the northern terrestrial biosphere. In contrast, APO is a tracer of surface ocean exchange with the largest seasonal variability observed over mid-to-high latitude oceans in both hemispheres. APO offers a distinct perspective for studying atmospheric mixing within and above the marine boundary layer, the long-range tracer transport into and out of the remote Southern Hemisphere, and the ability for inverting tracer flux over the SO from atmospheric measurements.

Here we use output from the APO-MIP1 (Stephens et al., 2025), which generated a suite of forward ATM simulations of APO and its components (air-sea $O_2$, $CO_2$, and $N_2$ flux, and fossil fuel $CO_2$ emission and $O_2$ uptake) from different source fields. This effort was initially motivated by a need to support the calibration of hemispheric-scale seasonal air-sea APO flux estimates from spatially and temporally sparse observations from airborne campaigns (e.g., Jin et al., 2023), stations, and ships. Here we focus on the other goals of APO-MIP1 which were to use atmospheric APO observations to characterize errors in ATMs and APO flux products.



In Section 3.1, we describe APO measurements from surface stations, aircraft, and ships, and the experimental design of APO-MIP1 using eight ATMs to simulate transport of three ocean APO flux products, and two fossil fuel products. In Section 3.2, we evaluate simulations against observations, revealing large model spread and errors at eastern Pacific surface stations due to mixing uncertainties, while airborne column-average data show smaller cross-ATMs variability and errors. In Section 3.3, we analyze diabatic mixing rates, demonstrating that ATMs generally overestimate mid-latitude mixing in both hemispheres, allowing us to separate transport and flux-related biases. In Section 3.4, we examine simulations of shipboard data around Drake Passage and the Antarctic Peninsula, revealing that current ATMs and flux products underestimate meridional gradients in APO seasonal amplitude from 53-65°S. The models also fail to capture the APO contrast between Palmer Station flask samples and nearby in-situ ship data due to limitations in representing local topographic flows with coarse-resolution ATMs.

## 2. Materials and Methods

### 2.1 Definition of APO

APO (per meg) is calculated from atmospheric observations of relative changes in the $O_2/N_2$ ratio (per meg) and $CO_2$ mole fraction (ppm) according to Stephens et al. (1998) as

$$APO = \delta(O_2/N_2) + \frac{1.1}{X_{O_2}}(CO_2 - 350),\tag{1}$$

with

$$\delta(O_2/N_2) = \left(\frac{\left(\frac{O_2}{N_2}\right)_{sample}}{\left(\frac{O_2}{N_2}\right)_{reference}} - 1\right)\cdot 10^6.\tag{2}$$

The factor 1.1 represents the approximate exchange ratio of $O_2$ to $CO_2$ in terrestrial biospheric processes (Severinghaus, 1995). We note that this ratio generally varies from 1.01 to 1.14 in aboveground carbon pools across different temporal and spatial scales (Gallagher et al., 2017; Hockaday et al., 2009; Keeling, 1988; Worrall et al., 2013). This ratio also exhibits diurnal change and varies between respiration and photosynthesis in biosphere-atmosphere $O_2$ and $CO_2$ exchanges (Faassen et al., 2023, 2024). With our focus on seasonal variations, we use 1.1 as



representative of the $O_2$ to $CO_2$ exchange ratio during seasonal growth and decay of terrestrial
biota. A sensitivity test in Jin et al. (2023) showed that varying this ratio by ± 0.05 has only
minor effects on seasonal APO changes. $X_{O_2}$ (0.2094) is the reference dry-air mole fraction of $O_2$
used in the definition of the $O_2$ scale of the Scripps $O_2$ Program (Keeling et al., 2020). $\delta(O_2/N_2)$ is
expressed in units of per meg, while $CO_2$ is converted from ppm units to per meg units by
subtracting a reference value of 350 ppm and then dividing by $X_{O_2}$. APO observations are
typically expressed in per meg units, but they can be converted to ppm equivalent units by
multiplying by $X_{O_2}$.

## 2.2 Atmospheric measurements

The APO-MIP1 (Stephens et al., 2025) required model output sampled to match a collection of
surface station, airborne, and shipboard observations, and also accepted optional output at
additional locations, at higher time resolution, and for full 3-D fields, as shown in Tables S1-2.
Here we evaluate model APO simulations using observation data collected at 10 surface stations,
on 10 airborne campaigns from three projects, and one repeated shipboard transect from 50
cruises. We show sampling locations, and horizontal flight and ship tracks in Fig. 1. We use
surface station APO measurements (2009 to 2018) from 10 sampling sites mainly in the Pacific
from the Scripps $O_2$ Program surface flask network (Keeling & Manning, 2014; Manning &
Keeling, 2006). The airborne measurements (Stephens et al., 2021) were made on the NSF
NCAR GV aircraft during the HIAPER Pole-to-Pole Observation project from 2009 to 2011
(HIPPO, Wofsy, 2011) and the $O_2/N_2$ Ratio and $CO_2$ Airborne Southern Ocean Study in 2016
(ORCAS, Stephens et al., 2018), and from the NASA DC-8 aircraft during the Atmospheric
Tomography Mission from 2016-2018 (ATom, Thompson et al., 2022). Shipboard measurements
were made on transects crossing the Drake Passage by the NSF ARSV Laurence M. Gould from
2012-2017 (Stephens, 2025). Details of surface station, airborne, and shipboard APO
measurements are provided in Appendix A.
As the primary focus of this study is the APO seasonal cycle and its latitudinal distribution, we
remove interannual trends from the observational data. For surface station and airborne
measurements, we remove the long-term trend by subtracting a deseasonalized cubic spline fit



(smoothing parameter of 0.8) derived from the global mean APO time series using Scripps $O_2$
Program data following Hamme & Keeling (2008). For the ship data, we apply a similar
detrending procedure but use only South Pole Observatory (SPO) data to derive the long-term
trend.

**2.3 Components of APO in the atmosphere and prescribed surface fluxes**

APO exhibits seasonal variations primarily driven by air-sea exchange ($F_{APO}^{ocn}$), which comprises
three components: air-sea exchange of $O_2$ ($F_{O_2}^{ocn}$), $CO_2$ ($F_{CO_2}^{ocn}$), and $N_2$ ($F_{N_2}^{ocn}$). Additionally, APO is
influenced by fossil fuel emission of $CO_2$ ($F_{CO_2}^{ff}$) and consumption of $O_2$ ($F_{O_2}^{ff}$), which together
combine to form a sink for APO due to fossil fuel burning ($F_{APO}^{ff}$). Fluxes are defined as positive
to the atmosphere.
In this study, we primarily simulate APO by performing forward transport of these individual
flux components in ATMs, except one inverse model flux product that provides net $F_{APO}^{ocn}$ directly.
We combined these components to calculate the net atmospheric APO anomalies in units of per
meg as

$$\delta APO = \delta APO^{ocn} + \delta APO^{ff}, \quad (3)$$

with

$$\delta APO^{ocn} = \frac{1}{X_{O_2}} \cdot \Delta O_2^{ocn} - \frac{1}{X_{N_2}} \cdot \Delta N_2^{ocn} + \frac{1.1}{X_{O_2}} \cdot \Delta CO_2^{ocn}, \quad (4)$$

and

$$\delta APO^{ff} = \frac{1}{X_{O_2}} \cdot \Delta O_2^{ff} + \frac{1.1}{X_{O_2}} \cdot \Delta CO_2^{ff}. \quad (5)$$



where $\Delta O_2^{ocn}$, $\Delta N_2^{ocn}$, $\Delta CO_2^{ocn}$, $\Delta O_2^{ff}$, and $\Delta CO_2^{ff}$ represents the atmospheric fields in units of
deviations in ppm of each flux component ($F_{O_2}^{ocn}$, $F_{CO_2}^{ocn}$, $F_{N_2}^{ocn}$, $F_{O_2}^{ff}$, and $F_{CO_2}^{ff}$) that is forward
transport in the ATMs (Stephens et al., 1998). The δ sign denotes tracers in units of per meg.
We utilize three distinct ocean APO flux products: (1) the Jena product, which directly provides
$F_{APO}^{ocn}$ from an atmospheric APO inversion framework that assimilates surface station
measurements (Rödenbeck et al., 2008); 2) the CESM product, an Earth System Model
simulation with prognostic ocean biogeochemistry (Yeager et al., 2022; Long et al., 2021) that
generates separate flux components ($F_{O_2}^{ocn}$ and $F_{CO_2}^{ocn}$) ; and 3) the DISS product, which provides
separate observation-based flux components incorporates surface ocean dissolved oxygen
measurements (Garcia & Keeling, 2001; Resplandy et al., 2016) and $pCO_2$ data (Jersild et al.,
2017; Landschützer et al., 2016). $F_{N_2}^{ocn}$ for CESM and DISS is estimated by scaling ocean heat
fluxes from CESM and ERA-5, respectively, using the relationship of Keeling et al. (1993). For
fossil fuel contributions, we employ the OCO2MIP product for $CO_2$ emissions (Basu & Nassar,
2021) and the GridFED database for coupled $O_2$ and $CO_2$ fluxes from fossil fuel combustion
(Jones et al., 2021). Details of each product are provided in Appendix B. All flux fields were
linearly interpolated from their original temporal and spatial resolution to 1° longitude × 1°
latitude with daily temporal resolution from 1986 to 2020. When flux data were unavailable in
the earlier portion of this time period (Jena and OCO2MIP), we set the corresponding fluxes to
zero. Participating modelers were requested to simulate at least from 2009 to 2018, following
three years of spin up from 2006 to 2008, and optionally longer (Table 1). In addition to Jena,
which is simulated directly, we construct the two $\Delta APO^{ocn}$ products using Eq. 4 and two $\Delta APO^{ff}$
products using Eq. 5, as described in Appendix B. Fig. 2 illustrates the seasonal and latitudinal
flux patterns of these three ocean APO flux products and the fossil fuel APO flux from GridFed,
which serves as our primary fossil fuel flux dataset in this study.
**2.4 Atmospheric tracer transport models**
We simulate each component of APO in the atmosphere using the flux fields described in Section
2.3, and eight ATMs (see Table 1). All tracer atmospheric fields are modeled as tracer deviations





against an arbitrary background with concentrations in ppm dry air mole fraction (as for $CO_2$). These tracer mole fractions are later converted to deviations in units of per meg after subtracting the model-specific arbitrary reference according to Eq. 4. We describe key model parameters and setups below.

### 2.4.1 CAM-SD

The Community Atmosphere Model (CAM) version 6.0 is the atmospheric component of CESM2 (Danabasoglu et al., 2020). The version used here is run online with specified dynamics (SD), wherein the model is constrained with MERRA-2 reanalysis, and uncoupled from the other climate system components. Temperature and horizontal winds (u and v) are nudged to MERRA-2, 8 times per day, with a normalized strength coefficient of 0.25. Shallow convection is parameterized following the Cloud-Layers Unified by Binormals framework (CLUBB, Golaz et al., 2002), and deep convection is parameterized following Zhang & McFarlane (1995). CAM has not been used for tracer inversions, but has been evaluated extensively for its dynamical properties (e.g., Bailey et al., 2019; Kay et al., 2012)

### 2.4.2 CAMS_LMDZ

CAMS_LMDZ refers here to the offline transport model from the Atmospheric General Circulation Model of Laboratoire de Météorologie Dynamique, called LMDz. LMDz is the atmospheric component of the Earth System Model of Institut Pierre-Simon-Laplace (IPSL). It is also used to drive the offline model CAMS_LMDz, in which case its horizontal winds are nudged to those of the ERA5 reanalysis. From the computer code of LMDz, CAMS-LMDz only keeps the transport subroutines for advection (Hourdin & Armengaud, 1999), deep convection (Emanuel, 1991), thermals (Rio & Hourdin, 2008), and boundary-layer turbulence (Hourdin et al., 2006). All other processes are replaced by an archive of relevant meteorological variables (air mass fluxes, exchange coefficients, temperature, etc.) built with the full LMDz model at the target spatial resolution, thereby allowing relatively small computing time and resources for the offline model. LMDz ensures the physical consistency of the archive of meteorological variables. The meteorological variables are stored as 3-hourly averages. CAMS_LMDZ has been regularly participating in OCO-2 MIP (Byrne et al., 2023) and TransCom intercomparison studies.



### 2.4.3 CTE_TM5

TM5 is a tracer transport model used for simulating atmospheric trace gas chemistry and transport (Krol et al., 2005). We refer to it as CTE_TM5 because the model was run with the CarbonTracker-Europe (CTE) shell, but this does not alter the TM5 physics and chemistry. TM5 advection is computed using the slopes advection scheme (Russell & Lerner, 1981) and in this work it is driven by ERA-5 reanalysis wind fields (Hersbach et al., 2020), making it an offline model. The convection is computed from the convective entrainment and detrainment fluxes from the ERA-5 reanalysis. Free tropospheric diffusion is computed using the formulation by Louis (1979). Diffusion in the boundary layer is computed using the parametrization by Holtslag & Boville (1993), where the diurnal variability in the boundary layer height is computed using Vogelezang and Holtslag (1996). TM5 is widely used in inversions and regularly participates in MIPs, for different tracers at different model resolutions and driven with different wind reanalysis products (for example, Byrne et al., 2023; Friedlingstein et al., 2025; Gaubert et al., 2019; Krol et al., 2018).

### 2.4.4 TM3

TM3 (Heimann & Körner, 2003) is an offline atmospheric tracer transport model, in the present runs driven by meteorological fields from the NCEP reanalysis (Kalnay et al., 1996). It was run here on a spatial resolution of 5 degrees longitude, about 3.8 degrees latitude, and 19 vertical layers. The advection uses the slopes scheme (Russell & Lerner, 1981), which is the same as in TM5. Boundary layer mixing is parameterized according to Louis (1979). Vertical mixing due to sub-gridscale cumulus clouds is calculated using the mass flux scheme of Tiedke (1989). TM3 is the ATM used in Jena APO inversion (Rödenbeck et al., 2008), which is one of the flux products used in this study.

### 2.4.5 MIROC4-ACTM

MIROC4-ACTM is a new generation Model for Interdisciplinary Research on Climate (MIROC, version 4.0; Watanabe et al., 2008) atmospheric general circulation model (AGCM)-based chemistry-transport model (ACTM; Patra et al., 2018) . This AGCM is evolved from the Center for Climate System Research, University of Tokyo (CCSR) / National Institute for



Environmental Studies (NIES) / Frontier Research Center for Global Change, JAMSTEC (FRCGC) AGCM version 5.7b (Numaguti et al., 1997). The MIROC4 AGCM propagates only explicitly resolved gravity waves into the stratosphere through the implementation of a hybrid vertical coordinate system compared to its predecessor AGCM5.7b. The MIROC4 AGCM online-simulated horizontal winds and temperature are nudged to the Japanese 55-year Reanalysis (JRA-55) at 6-hourly time intervals (Kobayashi et al., 2015). MIROC4-ACTM produces "age-of-air" up to about 5 years in the tropical upper stratosphere (~1 hPa) and about 6 years in the polar middle stratosphere (~10 hPa), in agreement with observational estimates. The convective transport and inter-hemispheric transport of tracers in the model are validated using $^{222}$Radon and sulphur hexafluoride ($SF_6$), respectively (Patra et al., 2018).

### 2.4.6 NICAM-TM_gl5 and NICAM-TM_gl6

NICAM-TM is an atmospheric transport model based on the Nonhydrostatic Icosahedral Atmospheric Model (NICAM) (Niwa et al., 2011; Satoh et al., 2014). In this study, we used the offline mode of NICAM-TM, which uses air mass fluxes, vertical diffusion coefficients and other meteorological variables; those data are calculated in advance by an online calculation of NICAM, in which horizontal winds are nudged toward the JRA-55 data. In NICAM, the air mass fluxes are calculated consistently with the continuity equation while conserving tracer masses, which do not require any numerical mass fixing (Niwa et al., 2011). For APO-MIP1, two horizontal resolutions were used: "glevel-5" (gl5) and "glevel-6" (gl6), whose mean grid intervals are 223 and 112 km, respectively. The number of the vertical model layers is 40 and the top of the model domain is at approximately 45 km. The vertical diffusion coefficients are calculated with the MYNN (Mellor & Yamada, 1974; Nakanishi & Niino, 2004) Level 2 scheme (Noda et al., 2010). The cumulus parameterization scheme used in NICAM-TM is Chikira & Sugiyama (2010). Model performance for atmospheric constituent transport can be found in Niwa et al. (2011, 2012).

### 2.4.7 NIES

NIES-TM-FLEXPART is a coupled transport model combining Eulerian (NIES-TM) and Lagrangian (FLEXPART) models. It is a transport modeling component of the variational flux



inverse modeling system NIES-TM-FLEXPART-Variational (NTFVAR, Maksyutov et al., 2021).
The NIES Transport Model (NIES-TM) is an offline model, originally developed in the 1990s
(Maksyutov et al., 2008). In this study, the NIES-TM v.21 is used, which improves $SF_6$ transport
and tropopause height over the former v.08.1 (Belikov et al., 2013), as evaluated in Krol et al.
(2018), due to (a) using ERA5 hourly wind data, including vertical wind on model coordinates,
on 137 model levels and a 0.625° grid for preparation of the 4-hourly average mass fluxes on 42
hybrid-pressure levels, (b) transporting first-order moments (Russell & Lerner, 1981; Van Leer,
1977) for advection, (c) applying penetrative convection rate and turbulent diffusivity supplied
by the ERA5 reanalysis (Hersbach et al., 2020). The version v.21 is the same as used in the
OCO-2 MIP (Byrne et al., 2023). NIES-TM is coupled with the Lagrangian model FLEXPART
(Stohl et al., 2005) to provide refinement to the near field transport during the last 3 days prior to
the observation event as presented by (Belikov et al., 2016). FLEXPART model v.8.0 is driven
by 6-hourly JRA-55 winds, interpolated to 40 hybrid pressure levels and 1.25°x1.25° resolution.
The surface flux footprints are produced by FLEXPART at 1°x1° resolution and daily time step.
Table 1. Participating ATMs and model parameters.

| Abbreviation | Model System | Grid (latitude × longitude × levels) | Meteorology | Run start, valid period | Reference(s) |
|---|---|---|---|---|---|
| CAM-SD | Community Atmospheric Model | 0.9º × 1.25º × 56 | MERRA-2 | 1986, 1989-2019 | Danabasoglu et al., 2020 |
| CAMS_ LMDZ | Copernicus Atmosphere Monitoring Service | 1.875º × 3.75º × 39 | ERA5 | 1986, 1991-2020 | Chevallier, 2013; Chevallier et al., 2005, 2010 |
| CTE_TM5 | CarbonTracker Europe | 1º × 1º × 25 | ERA5 | 2000, 2003-2020 | Luijkx et al., 2017 |



| Jena_TM3 | TM3 | 4º × 5º × 19 | NCEP | 1986, 1989-2020 | Heimann & Körner, 2003 |
|---|---|---|---|---|---|
| MIROC4-ACTM | MIROC4-ACTM | 2.8º × 2.8º × 67 | JRA-55 | 1986, 1991-2020 | Chandra et al., 2022; Patra et al., 2018 |
| NICAM-TM_gl5 | NICAM-based Transport Model | ~223 km × 40 | JRA-55 | 1986, 1989-2020 | Niwa et al., 2011, 2017 |
| NICAM-TM_gl6 | | ~112 km × 40 | | | |
| NIES | NIES-TM-FLEXPART | 3.75º × 3.75º × 42 (NIES-TM); 1º × 1º × 40 (FLEXPART) | JRA-55 | 2000, 2003-2020 | Belikov et al., 2016; Maksyutov et al., 2021 |

## 2.5 Outputs from transport models

For each ATM, we required simulations for all species sampled to match with the observation locations and times in a subset of the full ObsPack $CO_2$ files GLOBALVIEWplus v7.0 ObsPack (Schuldt et al., 2021) , excluding the model spin-up period. This subset corresponds to existing APO observations that are analyzed in this study from Scripps $O_2$ Program surface stations, NSF NCAR airborne observations, and NSF NCAR and AIST/JMA shipboard programs. The full list of these records is in Table S1. We note that, while the HIPPO, ORCAS, ATom, and Gould ObsPack files contain $CO_2$ observations from different instruments, their 10-sec sampling times align with the NSF NCAR APO measurements, except during calibration periods for either instrument.

We also received optional output, which includes the full set of ObsPack files, 3-D atmospheric fields, meteorological variables, additional ship data, and output at additional fixed sites (Table S2). Further details are provided in the APO-MIP1 protocol available at Stephens et al. (2025). We obtained output matching the full set of ObsPack files from four ATMs, which will be useful for future network design. We obtained daily mean 3-D gridded concentration fields from six





ATMs. These fields support the calculation of diabatic mixing rates, which we use to evaluate
ATMs and the flux products, following the method of Jin et al. (2024). Details are in Section 3.2.
We also received hourly (from two versions of NICAM) or 3-hourly (from NIES) output for an
extensive list of sites with past or ongoing APO measurements, and co-located samples for ship
sampling programs of NIES VOS, AIST R/V Mirai, and UEA Cap San Lorenzo (Hamburg Süd)
from three models. These data are not analyzed in this study, but are made available at Stephens
et al. (2025).

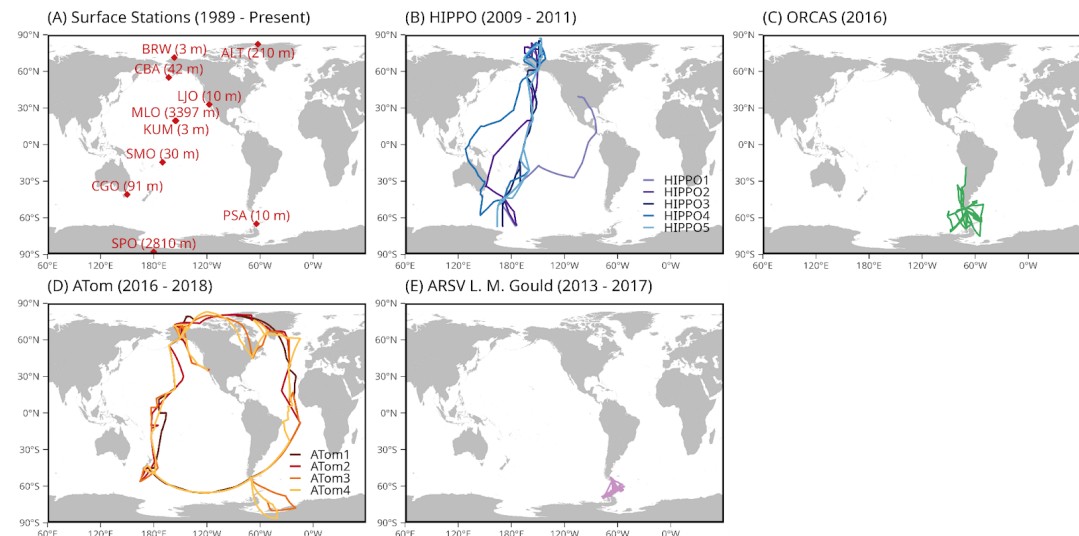


Figure 1: Geographic distribution of APO observations used in this study: (A) Scripps $O_2$
Program surface stations (red diamonds) with station codes and inlet elevation in meters above
sea level; (B) HIPPO (1 to 5) airborne campaign horizontal flight tracks covering the Pacific
Ocean; (C) ORCAS aircraft measurements concentrated in the Drake passage; (D) ATom (1 to 4)
airborne campaign horizontal flight tracks covering the Pacific and Atlantic Oceans; and (E)
Ship-based measurements from the RV *Laurence M. Gould* operating in the Drake passage.

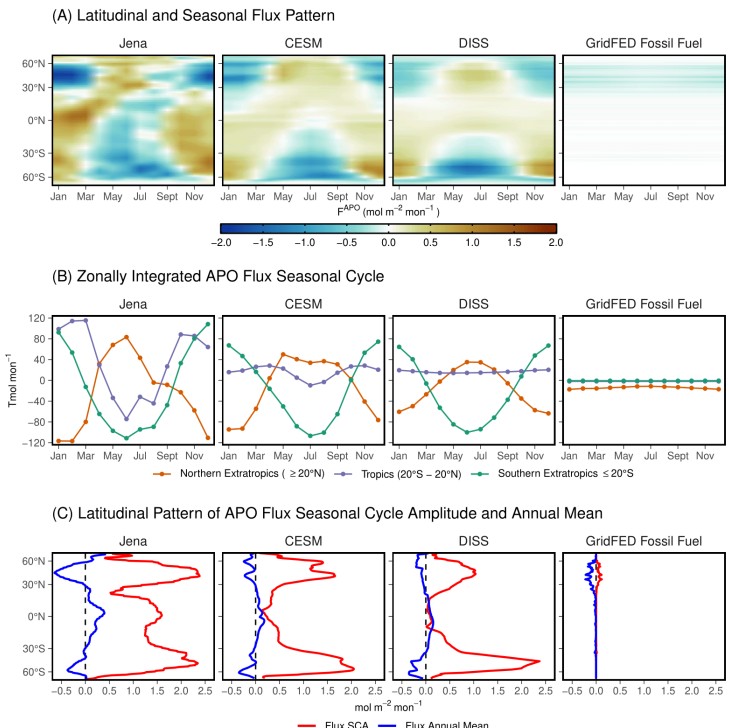


Figure 2: Comparison of APO flux patterns from the three air-sea flux products (Jena, CESM, and DISS) and fossil fuel emissions (GridFed), averaged from 2009 to 2018. (a) Hovmöller diagrams showing the spatiotemporal distribution of APO fluxes (mol m$^{-2}$ mon$^{-1}$) as a function of latitude and month. (b) Seasonal cycles of zonally integrated fluxes for three latitude bands: Northern Extratropics (≥20°N, orange), Tropics (20°S-20°N, lavender), and Southern Extratropics (<20°S, green). (c) Latitudinal profiles of flux seasonal cycle amplitude (SCA, red) and annual mean flux (blue). For the annual mean profiles (blue lines in panel C), only the latitudinal gradients should be interpreted, as the global means may contain biases in the ocean flux products, which are not the focus of this paper.





## 3. Results and discussion

## 3.1 APO model-observation comparisons at surface stations and along aircraft flight tracks

### 3.1.1 APO seasonal and latitudinal variations at surface stations

We show observations and model simulations of APO seasonal cycles at 10 surface stations of the Scripps $O_2$ program network in Fig. 3. We present annual mean values, seasonal cycle amplitudes (SCA), and phase from both observations and model simulations at these surface stations in Fig. 4, with model errors shown as colors. Observations show clear meridional gradients in APO annual means (Fig. 4A), with higher values in the Southern Hemisphere than Northern Hemisphere, and a southern tropical "bulge" (Battle et al., 2006; Gruber et al., 2001; Stephens et al., 1998). The APO SCA shows higher values in the high latitudes of both hemispheres, with larger amplitudes in the Southern Hemisphere compared to the Northern Hemisphere, yet reaches its maximum at the northern mid-latitude station Cold Bay (CBA) (Fig. 4B). The seasonal phase exhibits an approximately 6-month difference between hemispheres, while remaining relatively uniform within each hemisphere (Fig. 4C).

The higher annual mean APO in the Southern Hemisphere and the southern tropical "bulge" is a result of southward $O_2$ and $CO_2$ transport by the oceans, further amplified by net APO uptake in the Northern Hemisphere from fossil fuel burning (Keeling & Manning, 2014; Stephens et al., 1998). The larger APO SCA in mid- to high-latitudes reflects more pronounced seasonal flux cycles resulting from larger marine net primary production (NPP) and sea surface temperature changes in these regions. The thermal and biological effects on APO SCA are further enhanced at eastern Pacific coastal sites (e.g., LJO and CBA), where the shallow marine boundary layer traps high-APO air masses during summer. The 180-days phase difference between the two hemispheres is a result of different seasonal heating and cooling, as well as the biological cycle.

### 3.1.2 Biases in APO-MIP1 simulations at surface stations

APO-MIP1 simulations of APO annual means and seasonal cycles at surface stations broadly agree with observations (Figs. 3-4). Simulations driven by CESM fluxes show the best




agreement with observed APO features. For annual mean spatial patterns (Fig. 4A), CESM- and DISS-driven simulations show comparable performance in representing the southern tropical "bulge" and north-south gradient in annual means, while significantly outperforming simulations using the Jena flux model in northern stations. The main limitation of simulations using CESM fluxes is an overestimation of annual mean APO values across Pacific sites in the Southern Hemisphere, and an underestimation at LJO. Simulations using DISS fluxes also underestimate the annual mean APO at LJO.

APO SCA is well represented in simulations driven by CESM flux, but the SCA at LJO is significantly underestimated in all ATMs except CAM-SD. The underestimation is caused by an overly weak summer-time APO peak (Fig. 3), which also leads to the small annual mean presented above. Simulations using DISS flux generally underestimate SCA, especially in the high latitudes. Simulations using Jena flux, however, generally overestimate the SCA in the mid- to high-latitudes. We find largest SCA biases and cross-ATMs spread at LJO and CBA when using the Jena flux. The biases and model spread are closely related to underrepresentation in ATMs, and will be discussed in the next section. We note that the model biases and spread observed at surface stations are smaller than those reported in the previous TransCom-$O_2$ experiment (Blaine, 2005), indicating improved atmospheric transport modeling.

Phase simulations using CESM flux are consistent with observations at most stations, except at two northern low-latitude stations, KUM and MLO, where we find too late seasonal minimum day by up to two weeks. Simulations using DISS flux show even larger biases, with earlier seasonal minimum days at all southern and northern low-latitude stations.

### 3.1.3 Impact of ATM mixing biases

We find APO-MIP1 simulations have large model spread and biases at two northern mid-latitudes stations, LJO and CBA (Fig. 3), especially simulations using Jena fluxes. We note that the interdependence of transport models and fluxes in inversions can be seen for the Jena flux product simulations at LJO (Figs. 3-4). As expected, we see good agreement with observations for the Jena flux product transported by the same model used in the Jena APO inversion (Jena_TM3). However, all other ATMs overestimate summertime APO, and consequently SCA, for the Jena flux product at LJO, CBA, and BRW. All other ATMs also



simulate too negative wintertime APO at LJO. These biases suggest a stronger regional APO
source in the Jena flux product that could have resulted from too rapid dilution of surface flux
signals at LJO in both summer and winter.
Surface station simulations using CESM flux (Figs. 3-4) also reveal elevated model spread and
observation deviations at LJO and CBA. At LJO, all ATMs underestimate summertime APO, and
consequently SCA, implying too weak upwind outgassing fluxes. The relative magnitude of
simulated summer-time peaks for CESM at LJO and CBA maintains a consistent pattern across
different flux products, with CAM-SD consistently showing the highest values and Jena_TM3
the lowest, regardless of the flux product used, suggesting consistent biases in the ATMs.
This substantial cross-ATMs variability highlights the challenges in accurately representing
complex atmospheric vertical transport processes in regions where strong temperature inversions
and stratocumulus clouds significantly influence vertical mixing (Naegler et al., 2007; Nevison et
al., 2008). The Jena flux product, derived from an inversion that assimilates these station data,
relies on the TM3 tracer transport model (Rödenbeck et al., 2008). Previous studies indicate that
TM3 consistently overestimates vertical mixing over the Eastern Pacific, leading to larger
inverted seasonal fluxes to match station observations (Jin et al., 2023; Naegler et al., 2007). Our
analysis suggests that in comparison to Jena_TM3, vertical mixing is weaker in the two versions
of NICAM, CAM-SD, MIROC4-ACTM, and CTE_TM5, which show larger summer-time APO
anomalies at LJO and CBA. This pattern is consistent across the three flux products considered.
The larger model spread at northern coastal sites (e.g., LJO, CBA, and BRW) also highlights the
limitations of current coarse-resolution ATMs in representing horizontal coastal flows and
land-sea breezes. At LJO, samples are collected only during steady west wind (from the ocean)
conditions (Keeling et al., 1998). However, ATMs failed to capture the actual small-scale
atmospheric conditions associated with on-shore winds during episodic storm systems, which
leads to significant underestimation of oceanic influence (Keeling et al., 1998). APO, as a tracer
of air-sea gas exchange, is particularly sensitive to the dilution effects in coarse-resolution
models.



### 3.1.4 APO seasonal and latitudinal variations along flight tracks and biases in APO-MIP1

We present zonal averages of APO annual means, SCA, and seasonal minimum days derived from airborne data, grouped into 10-degree latitude and 100-mbar bands in Fig. 5A-C (full seasonal cycles in Fig. S1). We further calculate these three metrics as column-average (black) and at 900-mbar (blue) in Fig. 5D-F, where we also compare them with surface station data (shown as red points). The airborne data show patterns similar to those seen at surface stations but provide detailed vertical structures. The vertical profiles consistently show larger SCA at low altitudes, indicating that the main drivers of SCA are near the surface, while annual means and seasonal phases remain uniform across altitudes. Airborne column averages show increasing SCA and decreasing annual means from low to high latitudes, with similar SCA and annual mean values north of 50°N (Fig. 5D-E), whereas station observations show peaks in the mid-latitudes (LJO and CBA) due to high-APO air masses being trapped below the summer marine boundary layer. This trapping effect is also evident in airborne data interpolated to 900-mbar.

We also calculate APO annual means, SCA, and phases using aircraft simulations from APO-MIP1 (full seasonal cycles in Fig. S1) and compare simulated and observed column averages (1000-400 mbar average) in Fig. 6, with biases in column averages and vertical profiles shown in Figs. S2 and S3-5, respectively. Airborne observation-model comparisons complement those using surface station data. We find similar model biases to those seen in surface data, for example, larger SCA at northern high latitudes with the Jena flux product and smaller SCA at high latitudes with the DISS flux product. The airborne data also reveal three key biases that are not resolved at surface stations. Observations suggest a consistent near-zero annual mean APO in the Southern Hemisphere (south of 30°S), with a spike between 40° and 50°S. However, all three flux products show gradually decreasing annual mean APO south of 30°S, with CESM and DISS flux products showing a smaller spike in magnitude between 40° and 50°S. Simulations using CESM and DISS flux products show a larger SCA in the northern mid-latitudes (40 - 60°N). Additionally, simulations using the Jena flux product in the low northern latitudes show a seasonal minimum day similar to the Southern Hemisphere phase. This bias is caused by low-latitude flux features in the Jena inversion that largely replicate the Southern Hemisphere cycle, likely due to limited observational constraints in this region (Jin et al., 2023).





Our analysis demonstrates that global airborne measurements provide distinct advantages over
station data for evaluating large-scale flux patterns due to the reduced sensitivity of column
averages to boundary-layer ATM transport uncertainties. While surface stations show substantial
cross-model spread in simulated APO (Figs. 3-4), column-averaged airborne simulations (Fig. 6)
reveal remarkable consistency across ATMs when driven by the same flux product. This
consistency suggests that column-averaged measurements effectively integrate over local
transport features that often dominate surface observations. Here we establish CESM as the most
realistic flux product among the three products. The better agreement between observations and
CESM-driven simulations provides a more reliable baseline for isolating and quantifying
transport-related discrepancies in individual ATMs.



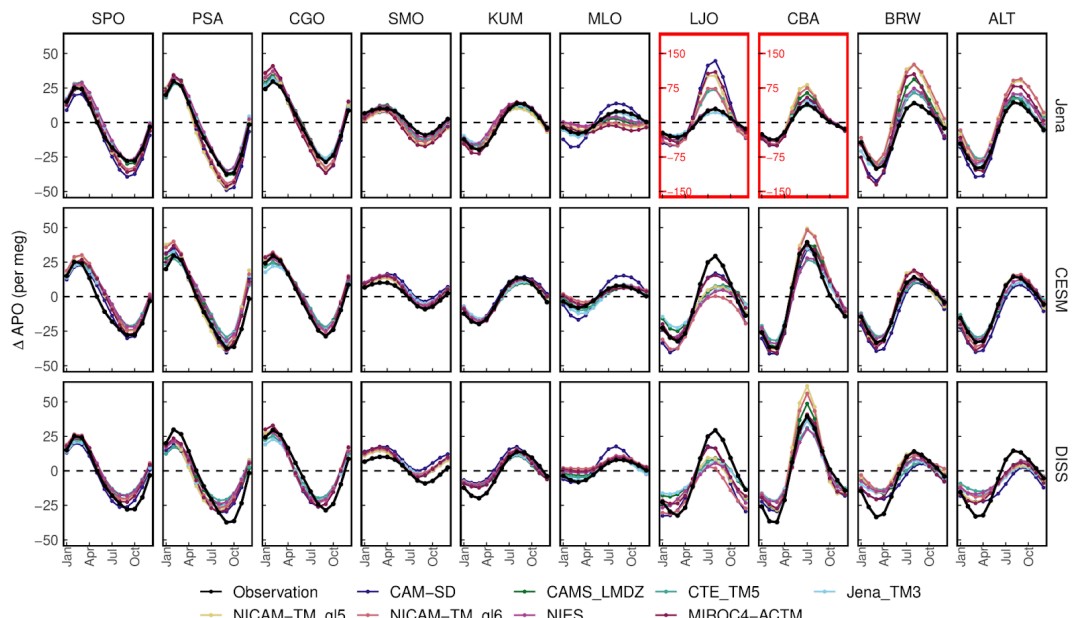

Figure 3: Comparison of simulated and observed APO seasonal cycles at 10 surface stations (Fig. 1A), organized from southern high-latitudes (left) to northern high-latitudes (right). In each panel, the black line represents observations, while colored lines show simulations from different transport models. Each row of panels corresponds to the three different flux products (Jena, CESM, and DISS). In each panel, the y-axis shows APO anomalies in per meg units, and the x-axis shows months from January to December. We note that, for LJO and CBA simulations using the Jena fluxes, a different y-axis range (three times larger) is used compared to the other panels. Observations and model simulations at each station are first detrended using a multiple-station weighted average trend. We calculate monthly mean seasonal APO from 2009 to 2018 for both observations and model simulations.



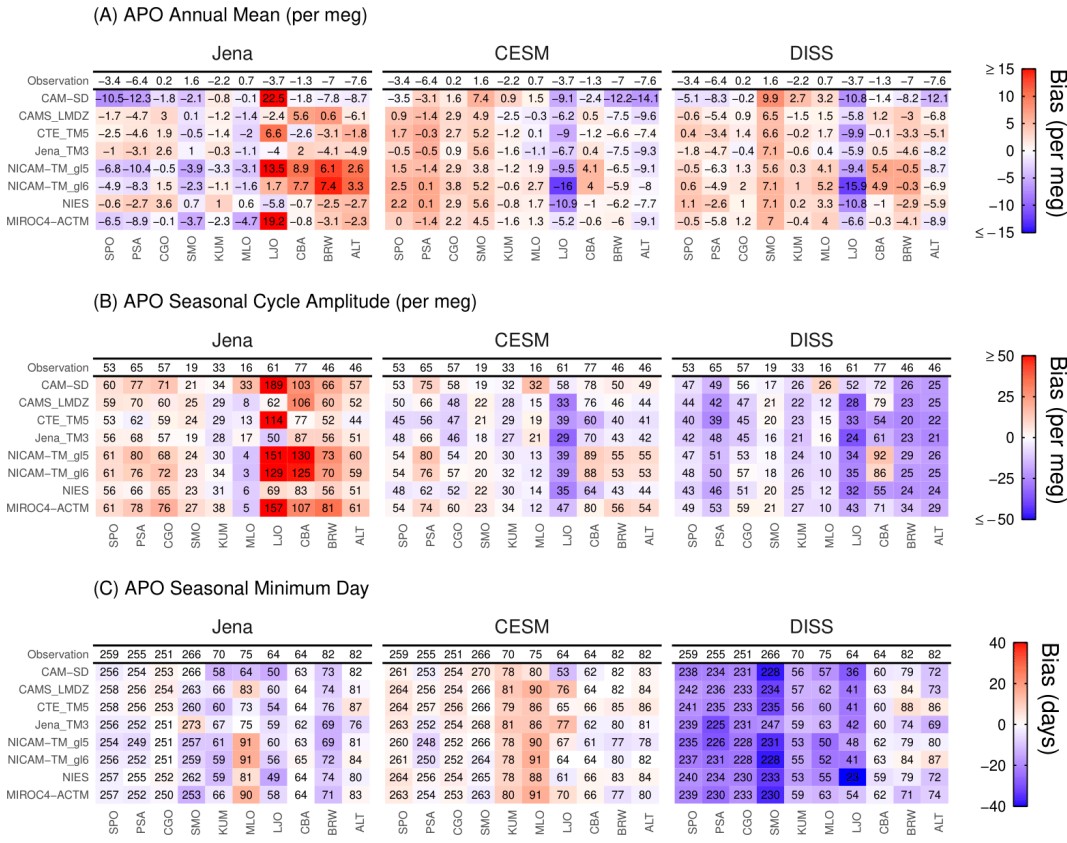

Figure 4: Evaluation of APO (A) annual mean relative to a multi-station global mean, (B) seasonal cycle amplitude, and (C) seasonal minimum day across surface stations using different flux-transport model combinations. For each panel, results are organized by flux products (JENA, CESM, DISS) in columns and transport models in rows, with observations on the top. The metrics are printed in black, with background colors indicating biases relative to observations. Positive bias is shown in red, and negative bias is shown in blue.



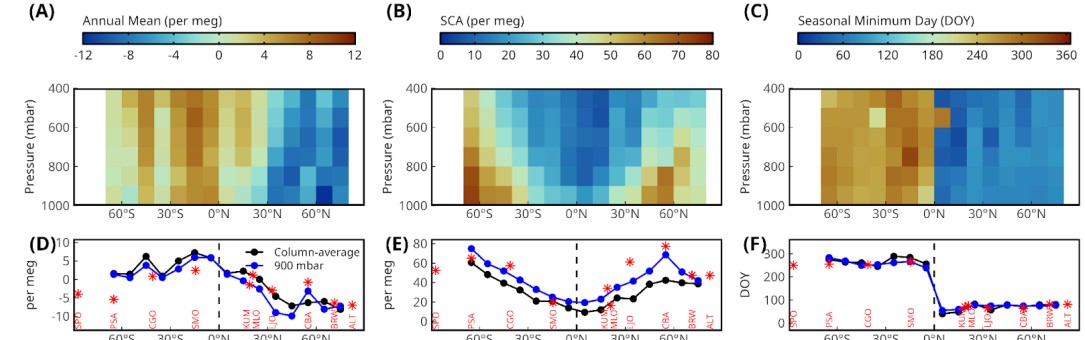

517

Figure 5: APO annual means (A and D), SCA (B and E), and seasonal minimum day (C and F) derived from airborne observations. In A-C, we show latitude-pressure distributions, with data binned into 10 deg latitude by 100-mbar pressure boxes. In D-F, we show 1000-400 mbar column-averaged (black) and 900-mbar interpolated (blue) values, and also surface station observations (2009 to 2018). Annual mean is derived from a two-harmonic fit with constant offset, where the global multi-station trend has been subtracted to detrend the airborne observations and center the values around zero globally. SCA is calculated as the peak-to-trough amplitude of the two-harmonic fit, and seasonal minimum day is calculated as the day of seasonal trough of the two-harmonic fit.



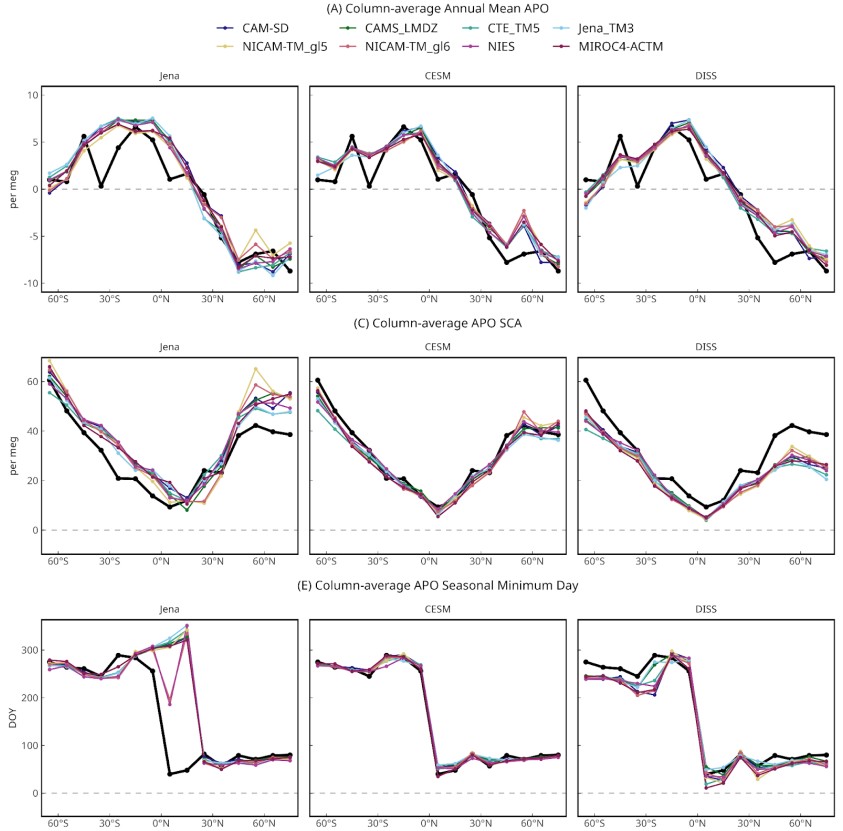

527

Figure 6: Comparison of column-average (1000-400 mbar) APO features across latitude from aircraft observations and model simulations using three different flux products (Jena, CESM, and DISS). The figure is organized into three sets of panels showing (A) annual mean APO relative to a multi-station global mean, (B) SCA, and (C) seasonal minimum day. For each feature, we show latitudinal distributions of observations (black lines) and model simulations (colored lines). We note that the global mean value has been subtracted from the annual mean values (A) at each latitude to highlight spatial patterns. We show the column-average (400-1000 mbar) seasonal cycles of observed and simulated APO for each 10° latitude band in Fig. S1.

## 3.2. Evaluation of diabatic mixing rates diagnosed from transport models

In this section, we evaluate the mixing timescale across mid-latitude moist isentropes of each ATM using the framework developed in Jin et al. (2024). This framework was applied to identify



biases in four ATMs in the mid-latitude Southern Hemisphere using two independent constraints: (1) diagnosed diabatic mixing rates, and (2) cross-isentrope $CO_2$ gradients. Here we extend the framework to use APO gradients, to include two more reanalysis products, and the analysis in the Northern Hemisphere. We evaluate six of the eight ATMs participating in APO-MIP1 that provide 3-D atmospheric fields (CAM-SD, CTE_TM5, Jena_TM3, NICAM-TM_gl5, NICAM-TM_gl6, and MIROC4-ACTM), which are required to diagnose diabatic mixing rates. Diabatic mixing rates and APO gradients are diagnosed based on the mass-indexed isentropic coordinate $M_{\theta e}$, which was first introduced by Jin et al. (2021). For each pair of transport models and flux products, we resolve cross-$M_{\theta e}$ diabatic mixing rates and cross-$M_{\theta e}$ APO gradients in the mid-latitudes of both hemispheres. We use observation-based diabatic mixing constraints diagnosed from four meteorological reanalyses, and observed APO gradient constraints calculated from three airborne campaigns. The detailed methodology for calculating $M_{\theta e}$ surfaces, diabatic mixing rates, and cross-$M_{\theta e}$ APO gradients is provided in Appendix C.

We show the climatological monthly mean diabatic mixing rates of two $M_{\theta e}$ surfaces in the Southern Hemisphere in Fig. 7, as well as schematics of the geographic distribution of the two $M_{\theta e}$ surfaces. For each ATM, mixing rates in Fig. 7 are calculated from APO and averaged over three realizations diagnosed from using three flux products. The reanalysis mixing rates are calculated from moist static energy (MSE) budget and shown as average and $1\sigma$ spread over the four reanalysis products. The six ATMs and the reanalyses show diabatic mixing rates with clear seasonal cycles, suggesting more rapid mixing across isentropes in the austral winter than summer. ATMs generally overestimate diabatic mixing rates, especially in the summer and winter, when there are large cross-$M_{\theta e}$ APO gradients that lead to well-defined mixing rates. Among the six ATMs, CTE_TM5 and Jena_TM3 show too rapid mixing that is biased high in all seasons. The other four ATMs align better with reanalysis, but still show significant overestimation for most of the year. MIROC4-ACTM shows the best performance. These findings align with Jin et al. (2024), which previously identified that the southern hemisphere summer-time mixing rates are overestimated in ATMs used for $CO_2$ inversions, with consistent results for the three ATMs (MIROC4-ACTM, Jena_TM3, and CTE_TM5) being used in both studies.



We find that biases in diagnosed diabatic mixing rates correlate with biases in cross-$M_{\theta e}$ APO
gradients in each season, with stronger diabatic mixing leading to smaller APO gradients (Fig.
8). Fig. 8 shows the ATM-diagnosed diabatic mixing rates and simulated APO gradients (points)
across six transport models and three flux products at two $M_{\theta e}$ surfaces (30 and $45 \times 10^{16}$ kg $M_{\theta e}$)
for three selected 2-month periods in the Southern Hemisphere. The points suggest clear linear
relationships between diagnosed mixing rates and simulated APO gradients for each flux product
(shown as fit lines for each flux product). The linear relationships persist across all seasons and
$M_{\theta e}$ surfaces, though with varying slopes depending on the underlying fluxes (Fig. 8). ATMs
generally underestimate cross-$M_{\theta e}$ absolute APO gradients (i.e., a closer to zero gradient) at both
$M_{\theta e}$ surfaces, corresponding to the overestimation of diabatic mixing rates in these models. For
each flux product, biases in cross-$M_{\theta e}$ APO gradients are always larger in fast mixing ATMs
(e.g., Jena_TM3 and CTE_TM5) compared to slow mixing ATMs (e.g., two versions of
NICAM-TM, MIROC4-ACTM, and CAM-SD), with MIROC4-ACTM showing the best
agreement. For each transport model, the simulated gradient shows clear spread across different
flux products. The largest spread occurs in austral winter and spring (Fig. 8C-D), when
simulations with the DISS fluxes show much larger gradients compared to CESM or Jena fluxes.
We note that the direct comparison of simulated and observed gradients for individual models is
complicated by the interplay of ATM biases and flux product biases.
To evaluate flux products independently of transport model biases, we leverage both diabatic
mixing rates and APO gradients. For each flux product, the intersection between the mixing
rate-gradient linear fit and the MSE-diagnosed mixing rate indicates the expected APO gradient
with realistic mixing characteristics. Therefore, we can evaluate large-scale flux features in the
flux products by comparing this expected gradient to the observed gradient. Our analysis in Fig.
8 suggests that CESM is the most realistic flux product in the mid-latitude Southern Hemisphere
in all seasons. The expected CESM gradients (intersections of thin blue line and vertical gray
band) fall within the observation uncertainty range in all seasons and surfaces except austral
summer at the $30 \times 10^{16}$ kg $M_{\theta e}$ surface (Fig. 8A), which suggests a slight underestimation of
uptake in the CESM product. The expected gradients of the Jena flux product also generally fall
within the observation uncertainty range, but shows an even larger underestimation in Fig. 8A.
The expected gradients of the DISS flux product have large biases in the mid-latitude Southern
Hemisphere. The expected gradient is significantly larger in the austral winter (Fig. 8C-D), and



significantly smaller at the $30 \times 10^{16}$ kg $M_{\theta e}$ surface in austral summer (Fig. 8A) and austral
spring (Fig. 8E), suggesting seasonal biases in the flux pattern.
Biases in expected gradients relative to observed gradients result from errors in the magnitude
and spatial distribution of air-sea APO flux, specifically the difference in flux magnitudes
between regions north and south of the target $M_{\theta e}$ surface. For instance, a positive expected
gradient bias during austral summer at the $30 \times 10^{16}$ kg $M_{\theta e}$ surface (Fig. 8A) in the DISS product
could stem from underestimated outgassing in high southern latitudes, excessive outgassing in
lower latitudes, or both. In addition, a flux product could produce realistic expected gradients
despite underestimating absolute fluxes both north and south of the $M_{\theta e}$ surface if the difference
remains correct. Resolving these inherent ambiguities requires additional observational
constraints from surface stations, ships, and aircraft, which we addressed in Section 3.1.
While the focus of Jin et al. (2024) was on the mid-latitude Southern Hemisphere, we extend our
analysis of the mid-latitude diabatic mixing rates to the Northern Hemisphere at the $45 \times 10^{16}$ kg
$M_{\theta e}$ surface (Fig. 9). ATMs also generally overestimate diabatic mixing rates in the Northern
Hemisphere, except during summer (JJA). Whereas MSE-diagnosed mixing rates peak in
northern summer, ATM-diagnosed mixing rates have their seasonal minimum at this time. We
note that APO gradients in ATMs are close to zero during JJA, leading to poorly defined diabatic
mixing rates. We carry out the same transport model and flux product analyses in the Northern
Hemisphere in January to March (Fig. 10A) and August to October (Fig. 10B). MIROC4-ACTM
still demonstrates the closest agreement with reanalysis data in both seasons, and CTE_TM5
shows the largest mixing rate bias. We note that TM3 and TM5 are based on similar
parameterization schemes, but TM3 outperforms TM5. In both seasons, the expected gradients
inferred from CESM flux align with the airborne observations, while Jena and DISS
overestimate and underestimate expected gradients, respectively.
Our attempt to diagnose mixing rates in ATMs in the Northern Hemisphere mid-latitudes using
ocean tracers alone is partly limited by the predominantly land surface. We find both summer
and winter peaks in seasonal diabatic mixing rates in the northern mid-latitudes, driven by strong
convection. Over land, convection peaks in summer due to strong surface heating that creates
unstable atmospheric conditions. Over the ocean, however, convection peaks in winter due to
larger air-sea temperature differences. Our ATM-diagnosed mixing rates in the Northern





Hemisphere may not capture the summer peak because atmospheric mixing processes over land may not be adequately reflected in transport of air-sea APO flux signals, which occurs initially over the ocean. This limitation is particularly significant in the Northern Hemisphere, where zonal mixing is slower (2-4 weeks) due to topographic blocking and stationary wave patterns. We plan to diagnose the land and ocean contrast in atmospheric diabatic mixing in the next APO-MIP1 by also forward transporting land tracers (e.g., $CO_2$ sources/sinks from the land biosphere). Our method is more robust in the Southern Hemisphere mid-latitudes due to faster zonal mixing (1-2 weeks) and the predominantly ocean surface. We also note that the distinct thermal capacities of land and ocean in the Northern Hemisphere create more complex surface $M_{\theta e}$ outcrops with larger latitudinal shifts across seasons (Jin et al., 2021), as shown in Fig. 9C. We, however, account for these shifts in our analysis.

Our analysis reveals that the ATM-diagnosed diabatic mixing rate primarily reflects an intrinsic characteristic of the transport model, at least in the Southern Hemisphere, showing little sensitivity to the underlying flux pattern, tracers, and land-ocean differences, particularly in models with smaller mixing rates (i.e., two versions of NICAM-TM, MIROC4-ACTM, and CAM-SD). These four models demonstrate consistent mixing rates across different flux products (Figs. 8 and 10). This consistency is further supported by our analysis of diagnosed mixing rates for individual APO components ($\Delta O_2^{ocn}$, $\Delta N_2^{ocn}$, $\Delta CO_2^{ocn}$, $\Delta O_2^{ff}$, and $\Delta CO_2^{ff}$) transported by ATMs with smaller mixing rates, which yields similar mixing rates despite these tracers having distinct signs, seasonal patterns, and magnitudes (Fig. S6). However, ATMs with faster mixing rate (e.g., Jena_TM3 and CTE_TM5) show large variability both across flux products (Fig. 9-10) and across tracers (Fig. S6). Notably, these two models exhibit approximately 50% slower diagnosed mixing rates for the fossil fuel $CO_2$ tracer ($\Delta CO_2^{ff}$) compared to the other ocean flux tracers in the austral summer at the $30 \times 10^{16}$ kg $M_{\theta e}$ surface. We note that the fossil fuel $CO_2$ tracer has its main source in the Northern Hemisphere, and its mixing at the mid-latitude Southern Hemisphere preferentially occurs in the upper troposphere. In contrast, the air-sea flux tracers have significant sources/sinks over the Southern Ocean with rapid cross-isentrope mixing preferentially in the lower troposphere. This behavior suggests that these models simulate distinctly different mixing patterns between the planetary boundary layer (0-2 km) and the free troposphere. Specifically, these models appear to have excessive vertical mixing in the boundary



layer while maintaining more realistic transport in the free troposphere. Our method, however,
assumes a constant cross-$M_{\theta e}$ diabatic mixing rate over the entire $M_{\theta e}$ surface. The excessive
boundary layer mixing causes the diagnosed mixing rates in these models to be overly sensitive
to the specific vertical distribution of air-sea APO flux components.
Our evaluation of ATMs using simulations from APO-MIP1 advances the original framework of
Jin et al. (2024) in three key aspects. First, we expand the experimental design by increasing the
number of participating ATMs to six and employing three different flux fields with each ATM,
generating 18 model realizations. This comprehensive matrix of simulations enables a more
systematic evaluation of both transport and flux-related biases. We demonstrate how atmospheric
tracer observations can be leveraged to independently evaluate and distinguish between biases in
surface fluxes and atmospheric transport models. Second, we enhance the robustness of our
MSE-diagnosed mixing rate calculations by incorporating two additional reanalysis products and
computing mixing rates at the native high resolution of each reanalysis, rather than averaging to
a coarser grid before the calculation. One limitation in our method is that we only use $M_{\theta e}$
calculated from MERRA-2 for each of the transport models rather than using $M_{\theta e}$ calculated
from the individual transport model, which in principle can be done by interpolating the
temperature and humidity from parent reanalysis to the ATM grid. This limitation would lead to
slight inconsistency between the actual $M_{\theta e}$ in the model and the value we assigned to it.
However, the differences between $M_{\theta e}$ calculated from different reanalyses remain small and our
method ensures consistency in geography of each $M_{\theta e}$ surface (Jin et al., 2021).



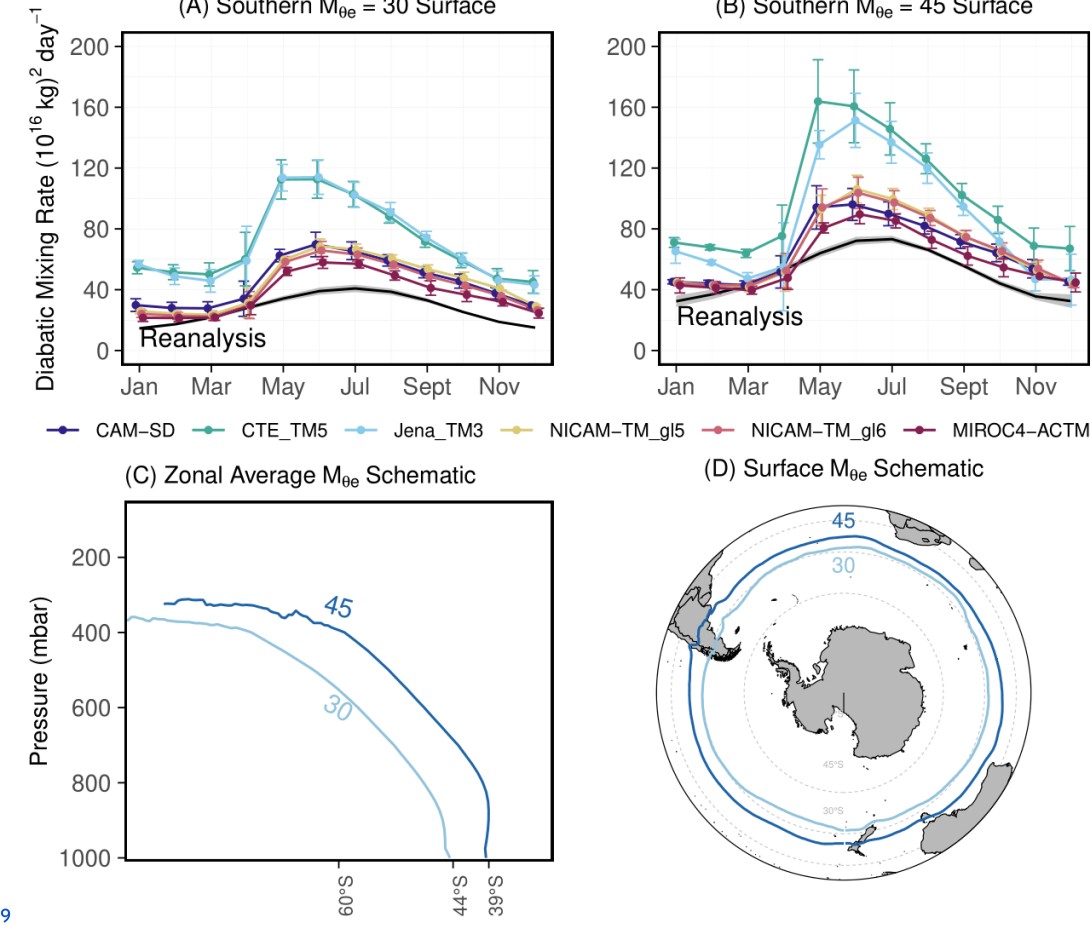

Figure 7: Climatological monthly diabatic mixing rates across the (A) 30 and (B) 45 ($10^{16}$ kg) $M_{\theta e}$ surfaces in the Southern Hemisphere. ATM-diagnosed mixing rates are derived from six ATMs in APO-MIP1 that provide 3-D APO fields. Error bars represent the 1σ spread across the 30 and 45 × $10^{16}$ kg $M_{\theta e}$ of three flux products used here. Black lines represent MSE-diagnosed mixing rates as the average of four reanalysis MSE budgets, while the gray shaded regions represent the 1-sigma spread. (C) Schematic showing latitude-pressure distribution of troposphere zonal annual average $M_{\theta e}$, and (D) annual average near-surface $M_{\theta e}$ contours of the 30 and 45 ($10^{16}$ kg) surfaces, computed from MERRA-2 reanalysis for the year 2009. These two $M_{\theta e}$ surfaces have very small seasonal meridional variability.



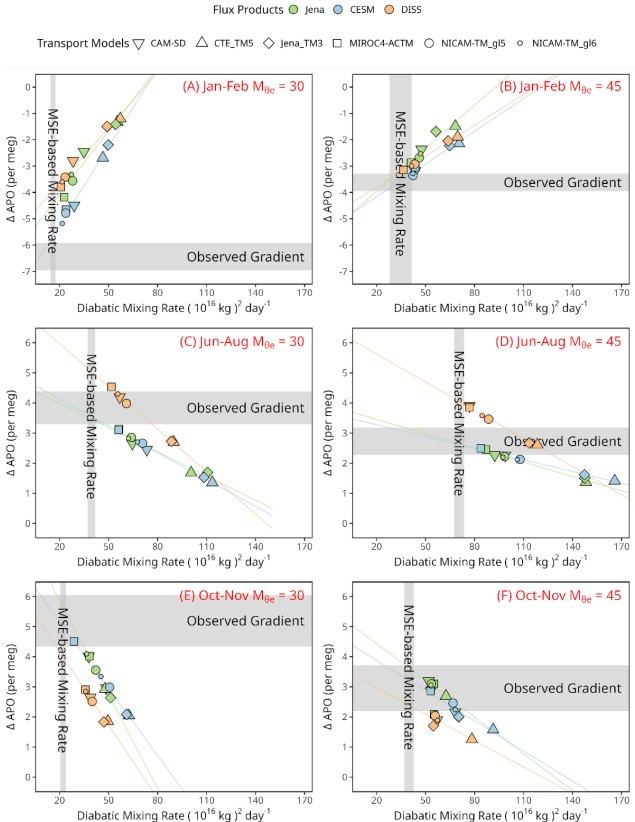

689

Figure 8: Using MSE-based diabatic mixing rates and airborne observations of cross-isentrope APO gradients to evaluate ATMs and flux models. Each panel compares model-diagnosed diabatic mixing rates (x-axis) and cross-$M_{\theta e}$ APO gradients (y-axis) at the $30 \times 10^{16}$ kg $M_{\theta e}$ surface (A, C, E, ~44°S surface outcrop) and at $45 \times 10^{16}$ kg $M_{\theta e}$ (B, D, F, ~39°S surface outcrop). Results are shown for three seasonal periods: Jan-Feb (a-b), Jun-Aug (c-d), and Oct-Nov (e-f) based on available airborne campaigns. Points represent individual model simulations, with colors indicating flux products (Jena, CESM, DISS) and symbols denoting different ATMs. Vertical gray bands show the 1σ range of MSE-based mixing rates derived from four reanalysis products. Horizontal gray bands indicate the 1σ range of observed APO gradients after spatial and temporal bias correction. Colored lines show linear fits of mixing rates and APO gradients for each flux product across different transport models.



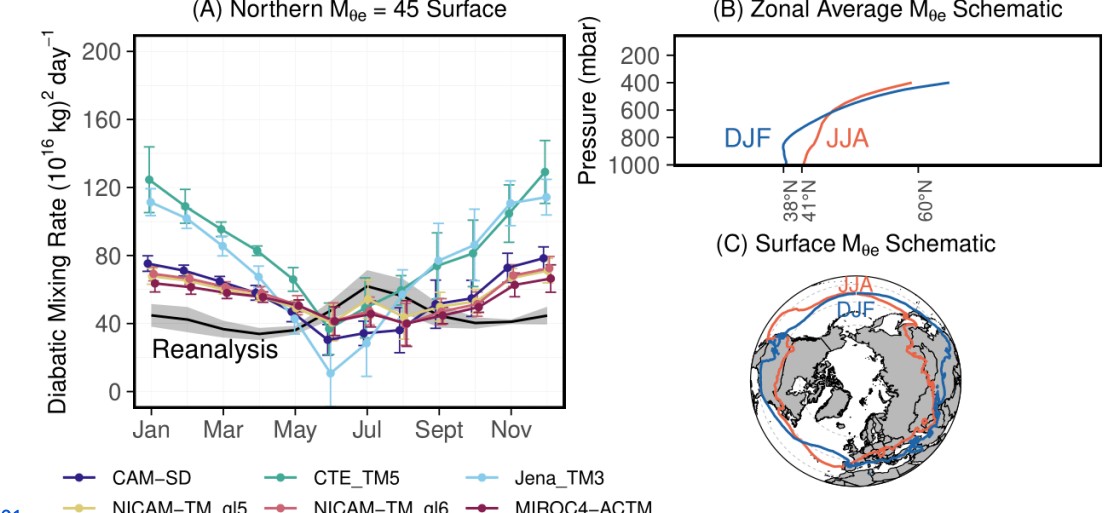

Figure 9: (A) Similar to Fig. 7, but showing climatological monthly diabatic mixing rates across the 45 ($10^{16}$ kg) $M_{\theta e}$ surface in the Northern Hemisphere. We note that JJA diabatic mixing rates in ATMs are poorly constrained due to close-to-zero cross-$M_{\theta e}$ APO gradients. (B) Latitude-pressure distribution of zonal average $45 \times 10^{16}$ kg $M_{\theta e}$ surfaces during boreal summer (JJA) and winter (DJF). The two $M_{\theta e}$ surfaces end at the tropopause, which is higher in the summer in the mid-latitudes. (C) Corresponding Earth surface outcrops of the JJA and DJF $45 \times 10^{16}$ kg $M_{\theta e}$ surfaces. Unlike in the Southern Hemisphere where seasonal meridional variations in $M_{\theta e}$ surfaces are small, the Northern Hemisphere shows pronounced seasonal shifts due to different land/ocean heating and cooling cycles.





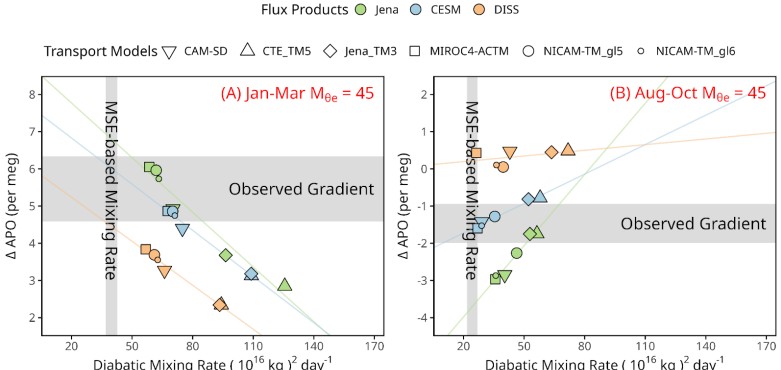


Figure 10: Similar to Fig. 8, but showing diabatic mixing rates and cross-$M_{\theta e}$ APO gradients in
the Northern Hemisphere late winter / early spring (A) and late-summer / early fall (B) of the 45
$\times$ $10^{16}$ kg $M_{\theta e}$ surface. We choose January to March and August to October due to sufficient
aircraft sampling and maximum cross-$M_{\theta e}$ APO gradients in these months.

## 716 3.3. Shipboard model-observation comparison over the Drake Passage

The APO-MIP1 simulations could not reproduce latitudinal variations in APO seasonal cycle
amplitude observed from shipboard measurements from 53 to 65°S over the Drake Passage and
adjacent to Tierra del Fuego and the Antarctic Peninsula. Observations reveal a strong
meridional SCA gradient (-2.1 per meg deg$^{-1}$, with deg positive northward), with SCA increasing
sharply towards higher southern latitudes (Fig. 11). Model simulations substantially
underestimate this latitudinal gradient (Fig. 11), showing weaker slopes averaged across ATMs
of -1.2 (Jena), -0.5 (CESM), and 0.8 (DISS) per meg deg$^{-1}$. Notably, these gradients remain
generally consistent across different ATMs for each flux product (±0.26, ±0.13, and ±0.29 per
meg deg$^{-1}$, respectively), suggesting this may predominantly be a result of zonal-scale latitudinal
biases in flux seasonality. Underrepresentation of enhanced summertime productivity along the
coast of the Antarctic Peninsula in flux products could also play a role. However, the Gould
typically only transits waters with elevated chlorophyll south of approximately 62°S while the
gradient biases appear further north. Furthermore, seasonally, the SCA biases are caused more by
underestimation of the winter/spring drawdown in APO at high latitudes, rather than the smaller
underestimation of summertime APO enhancement (Figs. S10-11). For CESM, this bias could



originate from incomplete process representation in the ocean biogeochemistry model and the
underestimation of winter mixed-layer depths in the Pacific sector of the Southern Ocean, which
has historically been a problem for Earth System Models (Sallée et al., 2013). The Jena flux
product provides the closest match to the observed SCA gradient. However, several limitations
remain, which likely stem from the coarse spatial resolution, limited atmospheric observational
constraints over the Southern Ocean, and underrepresentation of mixing patterns around the PSA
station (see details below and in SI). The DISS flux product is biased due to its underlying
assumptions and sparse observational constraints, as discussed in Jin et al. (2023).
Across ATMs, we find systematic differences of up to ±20% in simulated mean SCA for the
entire ship transects over the Drake Passage, independent of the input flux field, with CTE_TM5
consistently producing the smallest SCA and NICAM-TM_gl5 showing the largest. These
differences across ATMs are likely caused by differences in marine boundary-layer ventilation in
the models. Near-surface mixing over the Southern Ocean is challenging to model, owing to
complex boundary-layer structure, strong wind shear, frequent storm systems, SST variations,
and poorly represented clouds (Hyder et al., 2018; Knight et al., 2024; Lang et al., 2018; Truong
et al., 2020). The coarse-resolution models used here may struggle to capture such phenomena,
and the resulting variations in the concentration or dilution of flux signals near the surface drives
differences in mean APO SCA. The systematic spread also likely reflects biases in the
representation of large-scale diabatic mixing over the high southern latitudes. Models with strong
diabatic mixing rates, such as TM5, tend to dilute the meridional gradient of seasonal amplitude
through excessive mixing with lower-latitude air masses that have smaller SCAs, resulting in
reduced amplitudes at high southern latitudes.
We find that observed SCA at PSA (64.5°S) from SIO flask measurements (~ 70 per meg,
averaged from 2012 to 2017) is significantly smaller than nearby ship data from 64°S to 65°S (~
80 per meg). However, model simulations suggest similar values for both locations. The
shipboard measurements are closely tied to the SIO $O_2$ calibration scale, and any remaining scale
differences would be unlikely to affect the seasonal APO SCA. Rather, the observed SCA
difference occurs because SIO flask samples collected at PSA predominantly sample descending
air masses from the east that have passed over Anvers Island and the Antarctic Peninsula, with
peaks above 2000 m (characterized by small APO SCA), whereas the ship samples marine



boundary layer air including that over highly productive ocean regions (large APO SCA). As shown in Figs. S7-9, the SIO flasks are collected from the Terra Lab, on the east side of the station, with a wind selection criteria of 5-205°. Even while docked at Palmer (left-most points in Fig. 11), the Gould measurements show elevated SCA compared to PSA flask samples, because the pier is located to the west of the station with samples filtered to exclude air influenced by the station (Figs. S7-9). None of the ATMs, regardless of the flux product used, could reconstruct this feature, even though the models were sampled at the flask collection times. This difference is consistent with that seen between 900-mbar airborne samples and PSA flasks (Fig. 5E). The systematic bias points to the lack of resolution or physics that would be necessary, in either the reanalysis products or the ATMs, to accurately capture fine-scale circulation patterns, particularly the distinct air mass origins affecting ship versus station measurements. We note that the Jena flux product has been optimized to match seasonal APO cycles at Cape Grim Observatory (41°S) and at PSA (64.5°S), which may be the reason for its better performance on the SCA latitudinal gradient. It may do even better if the shipboard data were used in the inversion or if the effective sampling altitude of the SIO flasks at PSA were better accounted for.

Our analysis underscores the need for improvements in both ocean biogeochemistry models and ATMs. Future ocean process model developments should include improving accuracy of winter mixed-layer depths and higher-resolution ocean models with enhanced process representation to capture the fine-scale productivity patterns in the Southern Ocean. Additionally, current atmospheric transport models require improved resolution and physics to better represent the complex circulation patterns characteristic of coastal regions.





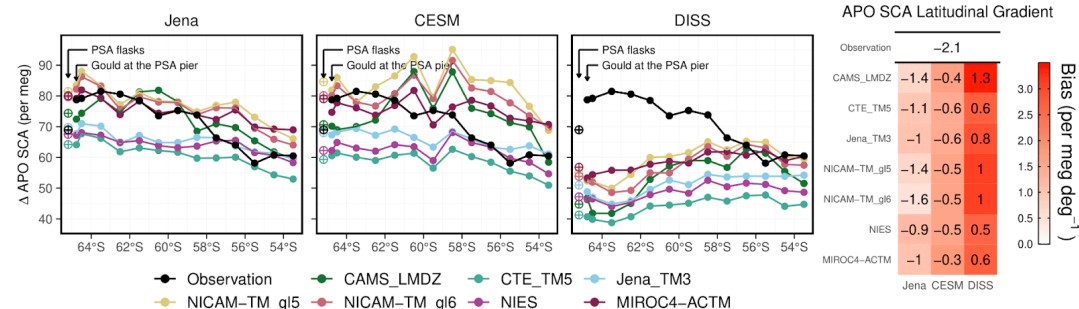

783

Figure 11: Latitudinal distribution of APO SCA across the Drake Passage region (53°S-65°S) derived from ship observations and model simulations. We calculate SCA by grouping observations and model simulations into 1 deg latitude bands, shown as points. Model results are color-coded by ATM and organized by flux products in separate panels. The full seasonal cycles of observed and simulated APO of these latitude bands are shown in Fig. S10. We also show SCA observed and simulated for the PSA flask record as open crossed circles (~64.5°S, shifted 0.7° south for visibility), and for ship data while the Gould is docked at or close to the PSA pier (left-most points, calculated by selecting data from 64.82°S to 64.72°S and 64.1°W to 64.0°W). The right-most three bands (53°S to 55°S) are typically downwind of Tierra del Fuego (Figs. S7-9). Both observational and model data for each latitude band or at PSA were detrended using corresponding cubic smooth spline fits from SPO. SCA was calculated using two-harmonic fits. The rightmost panel shows the SCA latitudinal gradients (per meg deg[-1]) from 53°S to 65°S, with red shading indicating model biases relative to observations. The gradient is calculated as linear fits of SCA from 53°S to 65°S for each ATM and flux product pair, and the observations. We exclude CAM-SD in this analysis because the ship data simulation is only available from 2012 to 2015 (i.e., missing 2016 to 2017 data).

## 3.4. Implications for APO and $CO_2$ inversions and ATM development

Our study motivates a community effort to conduct APO inversions. Estimates of spatial and temporal variations in APO fluxes can improve our understanding of ocean biogeochemical processes and heat transport, and support verification of fossil-fuel emission estimates (Pickers et al., 2022; Rödenbeck et al., 2023). Currently, only one global-scale APO inversion product from Jena CarboScope (Rödenbeck et al., 2008) exists. This product shows excessive seasonal flux



amplitudes (Fig. 2) in the tropics and northern mid-latitudes (~ 30 to 60°N) relative to the other
two flux products, which are more consistent with aircraft observations (Figs. 8 and 9). These
biases in Jena APO inversion partly result from limitations in the TM3 model, which exhibits
excessive vertical mixing, particularly in the eastern North Pacific, too rapid diabatic mixing in
the southern mid-latitudes, and underrepresentation of monsoon dynamics primarily due to
coarse resolution (Jin et al., 2023). The large spread and biases in ATMs shown in this study
highlight the importance of developing APO inversions using different ATMs and
methodologies, as this will improve our ability to fully assess methodological uncertainties and
potential biases in inverted air-sea APO flux estimates.
We encourage future inversion efforts to also assimilate column-mean data from airborne
campaigns, in addition to sparse surface stations, especially for studying climatological seasonal
fluxes. Our study finds that forward simulations from ATMs generally show large spread at
northeastern Pacific sites, particularly at LJO and CBA (Fig. 2), where simulations are sensitive
to model representation of the marine boundary layer and vertical mixing. The Scripps APO
observation network consists mainly of stations along a Pacific transect close to the primary
oceanic sources and sinks. Given this limited spatial coverage and our findings of significant
vertical mixing biases (e.g., at CBA and LJO) and local wind-direction biases (e.g., at LJO and
PSA) in ATMs at the station level, APO inversions that rely solely on these surface observations
may be subject to large representation errors. Airborne data, however, provide larger surface
footprints and column average metrics that are much less sensitive to vertical mixing biases. Our
analysis shows that ATMs are generally consistent with each other in simulating large-scale
annual and seasonal column-mean features along flight tracks (Fig. 6). Thus, inversions
configured to assimilate airborne column-mean observations would be promising. Further
improvement could also be achieved by incorporating shipboard observations to expand zonal
coverage, such as from the Gould, across the Atlantic (Pickers et al., 2017), and in the Western
Pacific (Tohjima et al., 2012). The study of Jin et al. (2023) used a different configuration of the
Jena inversion that also assimilated Japanese ship-based observations across the western Pacific
(Tohjima et al., 2012) from 40°S to 50°N. Forward transport of APO fluxes in that configuration
aligns better with station and airborne data compared to the configuration used in this study,
particularly in reducing the SCA bias in the tropics, suggesting better flux representations.



Biases in diabatic mixing diagnosed from ATMs (Section 3.2) imply that $CO_2$ inversions using these ATMs are also likely biased. A previous study showed that summer-time Southern Ocean $CO_2$ estimates from inversion products are correlated with corresponding simulated summer-time cross-isentrope $CO_2$ gradients in inversions (Long et al., 2021). The simulated gradients are shown to be biased too small due to too rapid diabatic mixing bias in ATMs leading to an overestimation of Southern Ocean $CO_2$ uptake in the summer (Jin et al., 2024). It is likely that biases in ATMs also contribute to the large spread found in OCO-2 MIP and Global Carbon Project (GCP) inversion ensembles (Byrne et al., 2023; Crowell et al., 2019; Friedlingstein et al., 2025; Peiro et al., 2022). We identify several priority areas for understanding biases in ATMs, particularly the inconsistency between diabatic mixing rates diagnosed from the MSE budgets of parent reanalysis and the tracer fields of coarser resolution ATMs identified here. These inconsistencies likely stem from several potential sources: (1) regridding of original reanalyses to the coarser resolution of the ATM grid, (2) for online GCMs using nudging, incomplete matching of the input meteorology, and (3) for offline models, recalculation or parameterization of convective mass fluxes in the coarser ATM. The first potential source of error from regridding could be evaluated by comparing MSE-based diabatic mixing rates from the parent and regridded fields as long as all components of MSE were included in the regridding. The second potential source of error from nudging could be evaluated by comparing MSE-based diabatic mixing rates from the regridded parent model and the nudged online simulation. Finally, the third potential source of error from recalculating or parameterizing vertical mass fluxes could be evaluated by comparing the MSE-based diabatic mixing rates from the regridded parent model and the tracer-based mixing rates from the ATM. It is notable that diabatic mixing rates diagnosed from two online models, MIROC4-ACTM and CAM-SD, which do not require regridding, are generally consistent with observations, with MIROC4-ACTM showing the best performance among all models (Figs. 7-10).

An important consideration is that the real atmosphere mixes MSE and tracers at different spatial and temporal scales. In the Northern Hemisphere, APO fluxes initially mix vertically over oceans, while strong $CO_2$ fluxes initially mix vertically over land. In contrast, MSE fluxes mix initially over both land and ocean. Due to the large land area in the Northern Hemisphere, the zonal mixing time scale is much longer (~ 2-4 weeks) so that diabatic mixing rates diagnosed from APO or $CO_2$ tracers could differ from each other and from those diagnosed from MSE



tracers. In the Southern Hemisphere mid-latitudes, these potential differences are much smaller due to the predominance of ocean and rapid zonal mixing (~ 1-2 weeks). In general, the timescales for diabatic mixing are longer than the timescales of zonal mixing, which support our approach of using tracer fluxes over both ocean and land to evaluate zonal-mean diabatic mixing. Future work should also develop metrics for quantifying along-isentrope (adiabatic) transport to complement our understanding of tracer mixing across isentropes. The timescales of adiabatic mixing influences tracer gradients along isentropic surfaces, which in turn affects diabatic mixing differently in the upper versus lower troposphere. It is also necessary to examine the sensitivity of mixing rates to model resolution, particularly vertical levels at the interface between the boundary layer and free troposphere, and boundary layer schemes. These ATM improvements are essential for enhancing both forward simulations and inverse estimates of surface fluxes.

## 4. Summary and Outlook

We conducted the Atmospheric Potential Oxygen forward Model Intercomparison Project (APO-MIP1) to generate forward simulations of APO and its components using different flux products and eight ATMs. This effort provides model APO simulations at surface stations, along aircraft flight paths, and on ships that can be directly compared with observations. Additionally, we provide 3-D APO fields from six of the eight ATMs. We use simulations from APO-MIP1 to evaluate eight ATMs and three flux products by comparing simulations against observations from surface stations, aircraft, and ships.

We find that model simulations of APO seasonal cycles using a given flux product show considerable summer-time spread at northern surface stations, particularly at two eastern Pacific stations, LJO and CBA (Fig. 3). The bias stems from challenges in accurately representing complex atmospheric vertical transport processes, marine boundary layer mixing, and coastal horizontal mixing in these regions. These findings highlight the limitations of current APO inversions that rely on a single ATM (i.e., TM3 used in Jena APO inversion) and sparse surface observations. However, model simulations of column-average APO resolved from sampling aircraft tracks are consistent across different ATMs, emphasizing the importance of airborne measurements for constraining large-scale flux features.



Using airborne observations and a moist-isentropic coordinate framework, we demonstrate that most ATMs overestimate diabatic mixing rates in the mid-latitudes of both hemispheres when compared to mixing rates derived from energy budgets of reanalyses. Among all ATMs used here, Jena_TM3 and CTE_TM5 show the largest biases. These constraints also enable us to separate flux biases from transport-related biases, allowing independent evaluation of flux models, which show that the CESM flux product is the best among the three flux products used in this study. This prognostic model outperforms two observation based products because of sparse atmospheric and surface observations, limitations in ATM used in atmospheric inversion, and because seasonal APO fluxes are driven by physical and biological processes that CESM represents well.

We encourage the broader community to develop new APO inversions, which could provide independent constraints on ocean biogeochemical processes and improve our understanding of the ocean carbon sink. Model simulations from APO-MIP1 can be used in other applications, including the calibration of methods for estimating seasonal air-sea APO fluxes from global atmospheric observations (e.g., Jin et al., 2023), constraining the representation of regional to global marine production in Earth system models (e.g., Nevison et al., 2012, 2015, 2018), and for understanding ESM biases in seasonal air-sea $CO_2$ exchange related to both thermal and non-thermal forcings. The transport simulations can also support the evaluation of long-term trends in $O_2$:$CO_2$ ratios over the Southern Ocean based on surface station gradients, useful for assessing biogeochemical responses to climate change.

We expect APO-MIP1 to continue evolving as an active collaboration examining atmospheric tracer transport and air-sea $O_2$ flux estimates. The current implementation excluded the air-sea $CO_2$ component and long-term flux trends from the Jena flux product, and does not include interannual and long-term flux trends in the DISS flux product, making these simulations unsuitable for interpreting interannual to long-term air-sea $O_2$ fluxes features. Thus, we only analyze APO seasonal cycles and meridional gradients here. The next phase of APO-MIP1 will address these limitations by incorporating updated inversion flux fields based on a larger set of atmospheric APO observations and including interannual variability. We will expand the scope by including terrestrial $O_2$ flux fields for $O_2$-specific analyses and seasonal-only component fluxes to investigate rectifier effects. Additionally, we plan to update air-sea $O_2$ fluxes derived





from surface ocean dissolved oxygen measurements by replacing Garcia and Keeling (2001) with fluxes calculated from recent machine learning interpolation of dissolved oxygen products (Gouretski et al., 2024; Ito et al., 2024; Sharp et al., 2023). We encourage broader participation from diverse modeling groups in the next phase of APO-MIP1.

## Appendix A: Surface station, airborne, and shipboard APO measurements.

The surface station APO observations from the Scripps $O_2$ program have been described in Keeling et al. (1998). Briefly, flask triplicates have been collected at biweekly to monthly frequency during clean background air conditions at a network of sites for over three decades, and returned to Scripps for analysis using interferometric and mass-spectrometric techniques. Here we use monthly data that was averaged from roughly bi-weekly data. The flask measurements are first adjusted to the middle of each month, parallel to the mean seasonal cycle for that station, before averaging. The APO-MIP1 output for these stations was reported matching the ObsPack $CO_2$ files from the Scripps $O_2$ Program, to take advantage of the established ObsPack format. These $CO_2$ measurements correspond to the same flask air on which $O_2$ is measured. The model output is treated in the same way as the observations to generate monthly means.

Airborne APO measurements from HIPPO, ORCAS, and ATom campaigns were made in situ with the NSF NCAR Airborne Oxygen Instrument (AO2), using a vacuum-ultraviolet absorption technique to measure $O_2$ and a single-cell infrared gas analyzer to measure $CO_2$ (Stephens et al., 2021). AO2 produces measurements every 2.5 s, which are averaged to 10 sec frequency for merging with other aircraft data. To correct for flight-specific sampling offsets, the in situ AO2 data were adjusted to agree with flask measurements collected during each flight using the NSF NCAR / Scripps Medusa flask sampler on a flight-by-flight average basis (Jin et al., 2023; Stephens et al., 2021).

HIPPO and ATom had nearly pole-to-pole coverage, and from near surface (150 - 300 m) to above the tropopause. HIPPO consisted of five campaigns between 2009 and 2011, and most data were collected above the Pacific. ATom consisted of four campaigns between 2016 and 2018, and each campaign had a Pacific transect and an Atlantic transect. ORCAS was a 6-week campaign with dense temporal sampling over the Drake Passage and ocean areas adjacent to the



tip of South America and the Antarctic Peninsula. The APO-MIP1 output for these aircraft measurements was reported matching the ObsPack $CO_2$ files for each campaign. These data are also at 10 sec frequency but correspond to different instruments with different calibration intervals. To match the observed and model time series, we mask observations when model output is not available, and vice versa. We also exclude any stratospheric data, with the stratosphere defined as water vapor concentrations below 50 ppm and either ozone concentrations exceeding 150 ppb, or detrended $N_2O$ levels (normalized to 2009) below 319 ppb (Jin et al., 2021). Water vapor and ozone were measured by the NOAA UAS Chromatograph for Atmospheric Trace Species instrument (Hintsa et al., 2021). $N_2O$ was measured by the Harvard Quantum Cascade Laser System instrument (Santoni et al., 2014). We filter the airborne data to exclude continental or urban boundary-layer air sampled while landing, taking off, or conducting missed approaches at airports (Jin et al., 2021).

Shipboard APO measurements from the ARSV *L. M. Gould* were made in situ during over 90 transects of Drake Passage on 50 cruises between 2012 and 2017 using a fuel-cell method for $O_2$ and a two-cell non-dispersive infrared gas analyzer for $CO_2$. The instrumentation was similar to a previously developed tower system (Stephens et al., 2003), but adapted and optimized for shipboard use. The instrument produces measurements at 1 min frequency. The cruises occurred in all months of the year but are more sparse during austral winter. The Gould operated almost exclusively between Punta Arenas, Chile and Palmer Station, Antarctica, in support of resupplying and transferring personnel to Palmer Station. The cruises span from 53° to 65°S in all months, and extend as far as 70°S during summer months. The APO-MIP1 output for the Gould was reported matching the ObsPack $CO_2$ file from the NOAA underway $pCO_2$ system. This system measures atmospheric $CO_2$ for 15 min every two hours. To match the observed and model time series, we first calculate hourly means for each and then mask observations when model output is not available, and vice versa.



## Appendix B: APO flux products

### B.1. Air-sea APO flux products

The first air-sea APO flux product (Jena) is air-sea APO flux from the Jena CarboScope APO Inversion (version ID: apo99X_v2021), which is available directly as $F_{APO}^{ocn}$ (update of Rödenbeck et al., 2008). In this inversion, the posterior fluxes (variable name: apoflux_ocean) were optimized to best match observed APO at 9 stations in the Scripps $O_2$ Program surface network (Manning & Keeling, 2006) and at 2 stations from the National Institute for Environmental Studies (Tohjima et al., 2012). The prior air-sea $CO_2$ flux was not included in the forward simulations here. We note that the exclusion of prior air-sea $CO_2$ flux has only minimal impact on the simulated APO seasonal cycle and north-to-south annual gradient but reduces the tropical "bulge" of annual mean by approximately 1 per meg and results in close to zero long-term APO trend. The Jena product is available from 1999 to 2020 originally with spatial resolution of 2° latitude × 2.5° longitude at daily intervals, converted to 1° × 1°. The Jena inversion used the TM3 transport model, which is also one of the models participating in APO-MIP1. In the case of TM3 forward transport simulation, the Jena inversion posterior fluxes have been re-run forward through the ATM, and thus this combination of fluxes and transport should agree well at the surface stations used for inversion optimization.

The second air-sea APO flux product (CESM) uses air-sea $O_2$, $CO_2$, and $N_2$ flux components from the Community Earth System Model (CESM2) Forced Ocean-Sea-Ice (FOSI) simulation (Yeager et al., 2022), which is forced by atmospheric fields from JRA55-do reanalysis (Tsujino et al., 2018) and prognostic ocean biogeochemistry using the Marine Biogeochemistry Library (MARBL, Long et al., 2021). The model directly produces $F_{O_2}^{ocn}$ and $F_{CO_2}^{ocn}$, while $F_{N_2}^{ocn}$ is calculated by scaling the ocean heat flux (Q, W m$^{-2}$) output using the relationship from Keeling and Shertz (1992) following

$$F_{N_2}^{ocn} = -\frac{1}{1.3} \cdot \frac{dS}{dT} \cdot \frac{Q}{C_p}, \tag{B1}$$

where dS/dT (mol kg$^{-1}$ C$^{-1}$) is the temperature derivative of solubility using solubility coefficients from Hamme & Emerson (2004). $C_p$ represents the specific heat capacity of



seawater, which is assumed to be 3993 J kg$^{-1}$ C$^{-1}$. The factor of 1/1.3 is to adjust the seasonal
amplitude due to the temporal lag between tracer flux and heat flux, as proposed by Jin et al.

1009 (2007).

These three CESM flux components have a resolution of 1° latitude × 1° longitude grid with the
North Pole displaced to Greenland. All fields are available from 1958 to 2020, but we only use
fluxes from 1986 to 2020. $F_{O_2}^{ocn}$ and $F_{CO_2}^{ocn}$ are output from the model at daily resolution, whereas
$F_{N_2}^{ocn}$ is calculated from monthly model heat fluxes then interpolated to daily resolution. This
version of CESM was designed to initialize a seasonal-to-multiyear large ensemble (SMYLE) of
coupled simulations for evaluating predictability. It is forced by observed meteorology starting in
1958, at which point it branches off of a FOSI configuration using JRA55-do atmospheric fields
as surface boundary conditions (Yeager et al., 2022). The FOSI simulation consists of six
consecutive cycles of 1958-2018 forcing, with the sixth cycle (used for SMYLE) extended
through 2020. Annual mean heat fluxes from this configuration show a small cooling drift over
the historical period, and thus the inferred annual mean and long-term trend of O$_2$ and N$_2$ flux
should not be interpreted as realistic.
The third air-sea APO flux product (DISS) uses bottom-up air-sea O$_2$ and CO$_2$ flux estimates
derived primarily from dissolved gas measurements. $F_{O_2}^{ocn}$ consists of a seasonal component
calculated from the dissolved O$_2$ measurement based climatology of Garcia & Keeling (2001),
with seasonal amplitude scaled by 0.82 according to Naegler et al. (2006), and an annual mean
component from the ocean inversion of Resplandy et al. (2016) for 21 regions using transport
from MITgcm-ECCO. The seasonal component (1.125° × 1.125° × monthly) was linearly
regridded to 1° × 1° × daily resolution. For the annual mean component, the original regional
values (21 regions) were spatially interpolated to 1° × 1° resolution while conserving the total
sum within each region, then temporally interpolated to daily values. We use $F_{CO_2}^{ocn}$ from the
machine learning interpolation of pCO$_2$ based air-sea CO$_2$ fluxes (Jersild et al., 2017;
Landschützer et al., 2016). The version of this product that we used provides fluxes from 1982 to
2020, with resolution of 1° latitude × 1° longitude × monthly, which we interpolated to daily. We





use Eq. B1 to calculate $F_{N_2}^{ocn}$ with heat fluxes from ERA5 reanalyses (Hersbach et al., 2020),
which is available from 1979 onwards, with resolution of 0.25° latitude × 0.25° longitude ×
monthly. Sea-surface temperature (SST) estimates required to calculate dS/dT (Eq. B1) are from
World Ocean Atlas (WOA) v2018 with resolution of 1° latitude × 1° longitude × monthly. SST is
available as a 1981 to 2010 climatology but we use it repeatedly for 1986 to 2020.
**B.2. Fossil fuel APO uptake products**
We used two products for $F_{APO}^{ff}$. The first product (GridFED) uses fossil $CO_2$ emission and $O_2$
uptake fluxes from Jones et al. (2021), downloaded from Jones et al. (2022). This product is
available from 1959 to 2020, with resolution of 0.1° latitude × 0.1° longitude × monthly, which
we interpolate to daily.
The second product (OCO2MIP) use $F_{CO_2}^{ff}$ as prepared for the OCO-2 Model Intercomparison
Project (MIP) version 10, downloaded from Basu & Nassar (2021), with resolution of 1° latitude
x 1° longitude x hourly. This $F_{CO_2}^{ff}$ product uses fossil fuel $CO_2$ emission from ODIAC (Oda et
al., 2018) for 2000 to 2019. For 2020, the flux was scaled from 2019 using the ratio of 2020 to
2019 global emissions reported by Liu et al. (2020). $F_{O_2}^{ff}$ is not available from this product, but
we scale the atmospheric field of $\Delta CO_2^{ff}$ by a factor of -1.4 to estimate $\Delta O_2^{ff}$ (Keeling, 1988;
Steinbach et al., 2011). We primarily use GridFED, except for CAMS_LMDZ where we use
OCO2MIP instead, because $F_{O_2}^{ff}$ from GridFED is missing for years after 2015. The differences
between these two products are negligible compared to the magnitude of ocean-driven APO
variations, for the seasonal metrics considered here.
**Appendix C: Calculation of $M_{\theta e}$, cross-$M_{\theta e}$ diabatic mixing rates and APO**
**gradients**
The mass-indexed moist isentropic coordinate $M_{\theta e}$ is defined as the total dry air mass under a
specific moist isentropic surface ($\theta_e$) in the troposphere of a given hemisphere. Surfaces of





constant $M_{\theta e}$ are parallel to surfaces of constant $\theta_e$ but the relationship changes with season, as
the atmosphere warms and cools. $M_{\theta e}$ surfaces have air mass ($10^{16}$ kg) as the unit, and are
adjusted to conserve dry air mass below the surface at any instant in time. $M_{\theta e}$ is calculated as a
function of $\theta_e$ and time following
$$M_{\theta e}(x,t) \;=\; \sum M_x(t)|\theta_{e_x} < \theta_e, \tag{C1}$$

where x indicates an individual grid cell of the atmospheric field, $M_x(t)$ is the dry air mass of
each grid cell x at time t, and $\theta_{e_x}$ is the equivalent potential temperature of the grid cell. For a
given $\theta_e$ threshold, the corresponding $M_{\theta e}$ value is calculated by integrating the air mass of all
grid cells with $\theta_e$ value smaller than the threshold. We only integrate air mass in the troposphere,
which is defined here as potential vorticity unit (PVU) smaller than 2. At each time step, this
calculation yields a unique value of $M_{\theta e}$ for each value of $\theta_e$ as well as a 3-D field of atmospheric
$M_{\theta e}$. Following the spatial pattern of $\theta_e$, $M_{\theta e}$ values generally increase from low to high altitudes
and from poles to equator. We generate daily $M_{\theta e}$ fields using four different reanalysis products
(MERRA-2, JRA-55, JRA-3Q, and ERA5) at their native resolution, avoiding potential
information loss from grid interpolation (Gelaro et al., 2017; Hersbach et al., 2020; Kobayashi et
al., 2015; Kosaka et al., 2024).
The calculation of diabatic mixing rates in ATMs is based on a box model approach, which uses
$M_{\theta e}$ as boundaries. A schematic of the box model is available as Fig. 1 of Jin et al. (2024). The
box model invokes tracer air mass balance, which recognizes tracer inventory change ($M_i$, Tmol)
of each $M_{\theta e}$ box equal to the sum of surface fluxes ($F_i$, Tmol day$^{-1}$) and the diabatic transport
between boxes ($T_{i,i+1}$, Tmol day$^{-1}$, positive poleward). The transport term is considered as a
diffusive system, which is parameterized as the product of diabatic mixing rate across the $M_{\theta e}$
boundary ($D_{i,i+1}$, $(10^{16}$ kg$)^2$ day$^{-1}$) and the tracer concentration ($\chi_{i+1}$, Tmol tracer per kg air mass)
gradient between two boxes. The full mass balance follows
$$\frac{\partial M_i}{\partial t} = \begin{cases} F_i + T_{i,i+1} & \text{if } i = 1 \\ F_i + T_{i,i+1} - T_{i-1,i} & \text{if } i > 1 \end{cases}, \tag{C2}$$


with



$$T_{i,i+1} = D_{i,i+1} \cdot \frac{\chi_{i+1} - \chi_i}{\Delta M_{\theta e}}. \tag{C3}$$

In these equations, i is the number label of the box and is set to be 1 at the highest latitude, $\Delta M_{\theta e}$ is the distance in $M_{\theta e}$ coordinates between box centers, which for evenly spaced boxes as used here, is the same as the total air mass of each box. In this study, we set the range of each $M_{\theta e}$ box to be $15 \times 10^{16}$ kg air mass, and therefore $\Delta M_{\theta e}$ equals the same value. The diabatic mixing rate (D) can be expressed as

$$D_{i,i+1}(t) = \frac{\left[\sum_{i'=1}^{i'=i} \left(\frac{dM_{i'}(t)}{dt} - F_{i'}(t)\right)\right]}{[\chi_{i+1}(t) - \chi_i(t)]} \cdot \Delta M_{\theta_e}. \tag{C4}$$

This method effectively reconstructs large-scale tracer transport features (T) in ATMs, as demonstrated in Jin et al. (2024). We note that the diabatic mixing rate is a property of the corresponding $M_{\theta e}$ and is theoretically insensitive to the choice of box sizes. We calculate climatological monthly average (2009 to 2018) diabatic mixing rates for each of the six transport models using the 3-D APO fields from transporting each of the three flux products (Figs. 7 and 9). To assign $M_{\theta e}$ at the model grid locations and times for each ATM, we always use $M_{\theta e}$ from MERRA-2 interpolated to the ATM grid, to ensure spatial consistency. Using other reanalyses only leads to small (< 5%) differences in ATM-diagnosed diabatic mixing rates (Jin et al., 2024).

Independent observational constraints on ATM-diagnosed mixing rates are calculated from moist static energy (MSE) budgets of four meteorological reanalyses (Figs. 7 and 9). MSE is a measure of static energy that is conserved in adiabatic ascent/descent and during latent heat release due to condensation, and naturally aligns with surfaces of $\theta_e$ or $M_{\theta e}$. This diagnostic approach offers more robust mixing rate estimates than tracer-based methods in part because MSE maintains consistent, non-zero gradients at each reanalysis time step, unlike chemical tracers. Additionally, MSE-based mixing rates are directly diagnosed from reanalysis on the original grid, avoiding potential artifacts introduced when these fields are interpolated to coarser transport model grids, and any recalculation of vertical mass fluxes and subgrid-scale mixing parameterizations in ATMs.



The MSE-diagnosed mixing rate calculation adapts our tracer box model framework. In this adaptation, we replace tracer inventory ($M_i$, Tmol) by MSE ($S_i$, J), replace surface tracer flux ($F_i$, Tmol day$^{-1}$) by surface heat flux ($Q_i$, J day$^{-1}$), and add an additional term to account for atmospheric radiative energy balance ($R_i$, J day$^{-1}$), following

$$D_{i,i+1}(t) = \frac{\left[ \sum_{i'=1}^{i'=i} \left( \frac{dS_{i'}(t)}{dt} - Q_{i'}(t) - R_{i'}(t) \right) \right]}{\left[ \chi_{i+1}(t) - \chi_i(t) \right]} \cdot \Delta M_{\theta_e} \qquad (C5)$$

We note that the gradient on the denominator in Eq. C5 represents the MSE density gradient (J per kg air mass) across the $M_{\theta e}$ surface. The calculation of these terms requires air temperature, specific humidity, surface heat flux, including surface sensible and latent heat flux, and radiative imbalance from reanalysis. Further details on the process to diagnose mixing rate from both ATMs and reanalyses can be found in Jin et al. (2024).

The cross-$M_{\theta e}$ APO gradient was calculated using data grouped into two adjacent boxes in the $M_{\theta e}$ space, with box centers spanning $15 \times 10^{16}$ kg air mass across the target surface boundary. For each box, we calculate the average APO concentration by trapezoidal integration of detrended APO as a function of $M_{\theta e}$ and dividing by the $M_{\theta e}$ range (Jin et al., 2021). We carry out the calculation for each airborne campaign, using the observations, model flight track output, and 3-D model fields. Flight-track estimated cross-$M_{\theta e}$ APO gradients are not directly comparable to simulated gradients from full 3-D fields, due to spatial and temporal coverage biases in airborne observations. We correct for both biases in the APO airborne observations and model flight track output (detailed in Supplement Text S1).

## Code and Data Availability

The 10 components of air-sea APO flux and fossil fuel APO uptake products, and the output of ATM forward transport simulations of these 10 components, including ATM samples at surface stations, ship transects, aircraft measurements, and 3-D atmospheric fields, are available at https://doi.org/10.5065/f3pw-a676 (Stephens et al., 2025). APO observations at surface stations from the Scripps $O_2$ network are available at https://doi.org/10.6075/J0WS8RJR (Keeling, 2019). All HIPPO 10-s merge data are available from Wofsy, 2017. Here we use updated HIPPO AO2



data from (Stephens et al., 2021a, 2021b, 2021c, 2021d, 2021e). All ORCAS 10-s merge data are available at Stephens (2017). Here we use updated ORCAS AO2 data from Stephens et al. (2021f). All ATom 10-s merge data are available at https://doi.org/10.3334/ ORNLDAAC/1925 (Wofsy, 2021), including the version of AO2 data used here. $O_2$ and $CO_2$ measurements from ARSV Gould are available at https://doi.org/10.26023/FDDD-PC3X-4M0X (Stephens, 2025). Note that airborne $O_2/N_2$ data are all on the Scripps $O_2$ Program SIO2017 $O_2/N_2$ scale defined on March 16, 2020, surface station data are on the SIO2023 $O_2/N_2$ scale defined on August 30, 2024, and shipboard data are on the SIO2023 $O_2/N_2$ scale defined on August 30, 2024. Airborne $CO_2$ measurements are on the WMO X2007 $CO_2$ scale, while station and shipboard $CO_2$ data are on the WMO X2019 $CO_2$ scale. The use of different scales has only minor impacts on interpreting APO seasonal cycles and latitudinal gradients. Code used to produce input flux files and to post-process submitted ObsPack files is available at https://doi.org/10.5065/f3pw-a676 (Stephens et al., 2025).

## Acknowledgments

We would like to acknowledge the efforts of the full HIPPO, ORCAS, and ATom science teams and the pilots and crew of the NSF NCAR GV and NASA DC-8, as well as the NSF NCAR and NASA project managers, field support staff, and logistics experts. Atmospheric $O_2$ measurements on HIPPO were supported by NSF grants ATM-0628519 and ATM-0628388. ORCAS was supported by NSF grants PLR-1501993, PLR-1502301, PLR-1501997, and PLR-1501292. Atmospheric $O_2$ measurements on ATom 1 were supported by NSF grants AGS-1547626 and AGS-1547797. Atmospheric $O_2$ measurements on ATom 2-4 were supported by NSF AGS-1623745 and AGS-1623748. The recent atmospheric measurements of the Scripps $O_2$ program have been supported via funding from the NSF and the National Oceanographic and Atmospheric Administration (NOAA) under Grants OPP-1922922 and NA20OAR4320278, respectively. The atmospheric $O_2$ measurements from ARSV Laurence M. Gould were supported by NSF grants ANT-0944761, PLR-1341425, and PLR-1543511. For sharing $O_3$, $N_2O$, and $H_2O$ measurements, we thank Jim Elkins, Eric Hintsa, and Fred Moore for ATom-1 $N_2O$ data; Ru-Shan Gao and Ryan Spackman for HIPPO $O_3$ data; Ilann Bourgeois, Jeff Peischl, Tom Ryerson, and Chelsea Thompson for ATom $O_3$ data; Stuart Beaton, Minghui Diao, and Mark Zondlo for HIPPO and ORCAS $H_2O$ data; and Glenn Diskin and Joshua DiGangi for ATom $H_2O$



data. YJ would like to acknowledge the Advanced Study Program Postdoctoral Fellowship in the NSF National Center for Atmospheric Research. This material is based upon work supported by the NSF National Center for Atmospheric Research, which is a major facility sponsored by the U.S. National Science Foundation under Cooperative Agreement No. 1852977. The work of FC was granted access to the HPC resources of CCRT under the allocation CEA/DRF, and of TGCC under the allocation A0130102201 made by GENCI. NC and PKP are supported by the Environment Research and Technology Development Fund (grant no. JPMEERF24S12205) and Arctic Challenge for Sustainability II (ArCS-II) project (grant no. JPMXD1420318865). YN is supported by JSPS KAKENHI (grant no. JP22H05006, JP80282151) and the Environment Research and Technology Development Fund (grant no. JPMEERF24S12210). IL and JH were supported by the Netherlands Organisation for Scientific Research (grant no. VI.Vidi.213.143 and NWO-2023.003).

## Author Contributions

YJ and BS carried out the research and wrote the paper with input from all co-authors. YJ, BS, and MC designed the research. MC prepared input fluxes for the transport models. BS provided airborne and shipboard observation data. EM provided surface station and airborne observation data. YJ, FC, NC, JH, IL, SM, YN, PP, CR, and JV provided forward transport model simulations. All authors contributed to reviewing and editing the text.

## Competing Interests

The contact author has declared that none of the authors has any competing interests.

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
