# Peer review of "The Atmospheric Potential Oxygen forward Model Intercomparison Project"

_EGUsphere, 2025_

## Referee Comment (RC1)

**Review's comments**

**Manuscript Number:** egusphere-2025-1736
**Title:** The Atmospheric Potential Oxygen forward Model Intercomparison Project (APO-MIP1): Evaluating simulated atmospheric transport of air-sea gas exchange tracers and APO flux products

**Authors:** Y. Jin, B. B. Stephens, M.C. Long, N. Chandra, F. Chevalier, J.J.D. Hooghiem, I.T. Luijkx, S. Maksyutov, E.J. Morgan, Y. Niwa, P.K. Patra, C. Rödenbeck, J. Vance

The authors conducted an atmospheric transport model (ATM) intercomparison (TransCom) experiment by using APO as a transport tracer to evaluate the representations of the transport and fluxes. They used three air-sea APO fluxes and eight atmospheric transport models including the model variants to simulate spatiotemporal variations in APO. The annual means, seasonal amplitudes, and day of seasonal minimum of the simulated APO were compared to those of observed APO from the Scrips surface measurement network, several airborne meridional cross-section observation programs, and shipboard measurements made on transects crossing the Drake Passage. Intensive comparison revealed that there is large spread in the performance of the ATMs. Especially from the comparison of the global airborne measurements between observation and simulation, the authors succeeded in evaluating the validity of APO fluxes. Additionally, the authors, applying a previously developed framework to evaluate diabatic mixing rate to this study, investigated the simulation-based and observation based diabatic mixing rates and APO gradients. The result also clearly showed the ATM-related biases in the diabatic mixing rate and differences in the representation of APO fluxes.

Previous TransCom experiments, using atmospheric tracers like CO2 with strong land sources and/or sinks, did not necessarily evaluate the atmospheric mixing above the ocean. Since APO is a unique tracer to study air-sea gas exchange, this TransCom experiment can provide a new perspective on the evaluation of the mixing processes in the ATMs especially above the ocean. I found that the paper is excellent, well organized, and contains material that should be published in EGUsphere. I highly recommend the manuscript to be published with the minor corrections as suggested below.

**Specific comments:**

Page 1, line 25: Although air-sea $CO_2$ exchange seldomly affects APO seasonal cycle, it may be oversimplified to say that APO is a tracer of air-sea $O_2$ exchange. I think it would be better to reword the relevant wordings to "a unique tracer to study air-sea gas exchange" or something like that.

Page 3, line 84: There are two "Nevison et al., 2008" in Reference. ("Nevison et al., 2008" is also cited in line 442.)

Page 6, line 165-166: Is it possible to quantify the extent of the "minor effect" for the ±0.05 change in the $O_2$:C exchange ratio for the terrestrial biota.

Page 9, line 256: It would be better to change "the ERA5 reanalysis" to "the ERA5 reanalysis wind fields (Hersbach et al., 2020)". Additionally, the reference should be removed from line 270.

Page 13, line 339: "Schuldt et al., 2021" is not listed in Reference.

Page 16, line 382-385: I think that it's not easy to imagine the meridional gradients in the annual mean APO from Fig. 4A. It would be better to refer Fig. 5A here.

Page 16, line 389: In this study, the authors mentioned that the days of seasonal minimum are latitudinally uniform in each hemisphere. Is it the same for the seasonal maximum? As far as I know, onboard observation in the western Pacific revealed that there was a difference in the meridional distribution pattern of the phases between seasonal minimum and maximum (Tohjima et al., 2012). If the APO observations adopted in this study show different meridional distribution patterns between seasonal minimum and maximum, it would be better to describe it in the manuscript to avoid misunderstanding of the readers.

Page 482-483: I'm not sure what the sentence means. Figure 6B seems to show that the simulated SCAs based on CESM agree with the observation while those based on DISS are underestimated.

Page 24, Figure 6: Please check the titles of figures: "( C ) Column-average APO SCA" and "( E ) Column-average APO Seasonal Minimum Day". Additionally, it would be better to add black line indicating the observation in the upper figure legends.

Page 29, line 664: It seems that only two key aspects are described in the paragraph.

Page 31, line 694-695: "(a-b)", "(c-d)", and "(e-f)" should be "(A-B)", "(C-D)", and "(E-F)".

Page 37: line 807: I'm not sure why the authors indicate Figs. 8 and 9 here.

Page 40, line 925: I think it would be better to explain the "rectifier effect" in the manuscript or to add appropriate reference here.

Page 44, line 1025: Does the scaling factor of 0.82 partially explain the underestimation of the simulated SCA based on DISS flux? If it's true, should it be mentioned in the manuscript?

Page 44, line 1025: "Naegler et al. (2006)" should read "Naegler et al. (2007)".

Page 60, line 1410-1413: "Kenneth et al., 2021" is not cited in the manuscript.

---

## Author Comment (AC1)

We thank the two anonymous reviewers for their valuable comments and efforts. We provide an updated manuscript, including updated figures, that incorporates the suggested changes and address specific comments below. In the following, the comments of the reviewer are presented in **blue**. Our responses are in **black**. Changes in the text based on the tracked version are in **red** .

**Reviewer #1**

The authors conducted an atmospheric transport model (ATM) intercomparison (TransCom) experiment by using APO as a transport tracer to evaluate the representations of the transport and fluxes. They used three air-sea APO fluxes and eight atmospheric transport models including the model variants to simulate spatiotemporal variations in APO. The annual means, seasonal amplitudes, and day of seasonal minimum of the simulated APO were compared to those of observed APO from the Scrips surface measurement network, several airborne meridional cross-section observation programs, and shipboard measurements made on transects crossing the Drake Passage. Intensive comparison revealed that there is large spread in the performance of the ATMs. Especially from the comparison of the global airborne measurements between observation and simulation, the authors succeeded in evaluating the validity of APO fluxes. Additionally, the authors, applying a previously developed framework to evaluate diabatic mixing rate to this study, investigated the simulation-based and observation based diabatic mixing rates and APO gradients. The result also clearly showed the ATM-related biases in the diabatic mixing rate and differences in the representation of APO fluxes.

Previous TransCom experiments, using atmospheric tracers like CO2 with strong land sources and/or sinks, did not necessarily evaluate the atmospheric mixing above the ocean. Since APO is a unique tracer to study air-sea gas exchange, this TransCom experiment can provide a new perspective on the evaluation of the mixing processes in the ATMs especially above the ocean.

I found that the paper is excellent, well organized, and contains material that should be published in EGUsphere. I highly recommend the manuscript to be published with the minor corrections as suggested below.

Specific comments:

1. Page 1, line 25: Although air-sea CO2 exchange seldomly affects APO seasonal cycle, it may be oversimplified to say that APO is a tracer of air-sea O2 exchange. I think it would be better to reword the relevant wordings to "a unique tracer to study air-sea gas exchange" or something like that.

We edited this sentence as follows.

L25 - 27: Atmospheric Potential Oxygen (APO, defined as $O_2 + 1.1 \times CO_2$) is primarily a tracer of ocean biogeochemistry and fossil fuel burning. APO exhibits strong seasonal variability at mid-to-high latitudes, driven mainly by seasonal air-sea $O_2$ exchange

2. Page 3, line 84: There are two "Nevison et al., 2008" in Reference. ("Nevison et al., 2008" is also cited in line 442.)

We want to refer to the following paper at both places. The error was fixed.

Nevison, C. D., Mahowald, N. M., Doney, S. C., Lima, I. D., and Cassar, N.: Impact of variable air-sea $O_2$ and $CO_2$ fluxes on atmospheric potential oxygen (APO) and land-ocean carbon sink partitioning, Biogeosciences, 5, 875–889, https://doi.org/10.5194/bg-5-875-2008, 2008.

3. Page 6, line 165-166: Is it possible to quantify the extent of the "minor effect" for the ±0.05 change in the O2:C exchange ratio for the terrestrial biota.

We added the following sentence to quantify the impact of the ±0.05 change in the O2:C exchange ratio.

L172 - 174: A sensitivity test in Jin et al. 2023 showed that varying this ratio by ± 0.05 only leads to ±5.1% changes in hemispheric average APO. The impact on APO seasonal cycle amplitude (SCA) is ±1.44% and ±0.41% in the Northern and Southern Hemisphere, respectively.

4. Page 9, line 256: It would be better to change "the ERA5 reanalysis" to "the ERA5 reanalysis wind fields (Hersbach et al., 2020)". Additionally, the reference should be removed from line 270.

 Changed as suggested.

5. Page 13, line 339: "Schuldt et al., 2021" is not listed in Reference.

The following reference was added.
Schuldt, K. N., Mund, J., Luijkx, I. T., Aalto, T., Abshire, J. B., Aikin, K., Andrews, A., Aoki, S., Apadula, F., Baier, B., Bakwin, P., Bartyzel, J., Bentz, G., Bergamaschi, P., Beyersdorf, A., Biermann, T., Biraud, S. C., Boenisch, H., Bowling, D., Brailsford, G., Chen, G., Chen, H., Chmura, L., Clark, S., Climadat, S., Colomb, A., Commane, R., Conil, S., Cox, A., Cristofanelli,

P., Cuevas, E., Curcoll, R., Daube, B., Davis, K., De Mazière, M., De Wekker, S., Della Coletta, J., Delmotte, M., DiGangi, J. P., Dlugokencky, E., Elkins, J. W., Emmenegger, L., Fang, S., Fischer, M. L., Forster, G., Frumau, A., Galkowski, M., Gatti, L. V., Gehrlein, T., Gerbig, C., Gheusi, F., Gloor, E., Gomez-Trueba, V., Goto, D., Griffis, T., Hammer, S., Hanson, C., Haszpra, L., Hatakka, J., Heimann, M., Heliasz, M., Hensen, A., Hermanssen, O., Hintsa, E., Holst, J., Ivakhov, V., Jaffe, D., Joubert, W., Karion, A., Kawa, S. R., Kazan, V., Keeling, R., Keronen, P., Kolari, P., Kominkova, K., Kort, E., Kozlova, E., Krummel, P., Kubistin, D., Labuschagne, C., Lam, D. H., Langenfelds, R., Laurent, O., Laurila, T., Lauvaux, T., Lavric, J., Law, B., Lee, O. S., Lee, J., Lehner, I., Leppert, R., Leuenberger, M., Levin, I., Levula, J., Lin, J., Lindauer, M., Loh, Z., Lopez, M., Machida, T., et al.: Multi-laboratory compilation of atmospheric carbon dioxide data for the period 1957-2020; obspack_co2_1_GLOBALVIEWplus_v7.0_2021-08-18, , https://doi.org/10.25925/20210801, 2021.

6. Page 16, line 382-385: I think that it's not easy to imagine the meridional gradients in the annual mean APO from Fig. 4A. It would be better to refer Fig. 5A here.

We now refer to Fig. 5A.

7. Page 16, line 389: In this study, the authors mentioned that the days of seasonal minimum are latitudinally uniform in each hemisphere. Is it the same for the seasonal maximum? As far as I know, onboard observation in the western Pacific revealed that there was a difference in the meridional distribution pattern of the phases between seasonal minimum and maximum (Tohjima et al., 2012). If the APO observations adopted in this study show different meridional distribution patterns between seasonal minimum and maximum, it would be better to describe it in the manuscript to avoid misunderstanding the readers.

Airborne observations suggest that the seasonal maximum date is also uniform in the mid- to high-latitudes of each hemisphere, with $1\sigma$ of 13 and 16 days in the Northern and Southern hemisphere, respectively, without a statistically significant trend. Ship data might be especially sensitive to near-field or within-basin ocean outgassing and uptake patterns, leading to obvious meridional gradients in the seasonal phase. Airborne column-mean data, in contrast, are more likely to represent large-scale to zonal-mean averages that average over smaller-scale zonal flux variations.

8. Page 482-483: I'm not sure what the sentence means. Figure 6B seems to show that the simulated SCAs based on CESM agree with the observation while those based on DISS are underestimated.

We intended to report errors in annual mean rather than SCA. We corrected this sentence below.

L493 - 495: Simulations using CESM and DISS flux products show larger annual mean values in the northern mid-latitudes (40 - 60°N).

9. Page 24, Figure 6: Please check the titles of figures: "( C ) Column-average APO SCA" and "( E ) Column-average APO Seasonal Minimum Day". Additionally, it would be better to add black line indicating the observation in the upper figure legends.

The labels are corrected as (A), (B), and (C). We also added a black line to indicate the observation in the figure legend.

10. Page 29, line 664: It seems that only two key aspects are described in the paragraph.

This was a typo. We corrected the sentence.

Corrected as suggested.

12. Page 37: line 807: I'm not sure why the authors indicate Figs. 8 and 9 here.

We intended to point to Fig. 6. We corrected this sentence as:

L819 - 822: This product shows excessive seasonal flux amplitudes (Fig. 2) in the southern low-latitudes (~ 30 to 0°S) and northern mid-latitudes (~ 30 to 60°N) relative to the other two flux products, which show better consistency with aircraft observations in their forward transport simulations (Fig. 6).

13. Page 40, line 925: I think it would be better to explain the "rectifier effect" in the manuscript or to add appropriate reference here.

We added the following sentence to expand on the rectifier effect.

L941 - 946: The seasonal rectifier effect refers to the creation of non-zero annual mean atmospheric concentration gradients at surface stations even with balanced seasonal $O_2$ fluxes. This occurs when fluxes correlate with seasonal variations in atmospheric mixing. For example, strong summer $O_2$ outgassing combined with shallow PBL heights concentrates APO near the surface, while higher winter PBL dilutes the $O_2$ uptake signal, resulting in observed annual mean APO gradients even when the annual mean flux is zero.

14. Page 44, line 1025: Does the scaling factor of 0.82 partially explain the underestimation of the simulated SCA based on DISS flux? If it's true, should it be mentioned in the manuscript?

We added the following sentence.
L1053-1058: Bent (2014) reported that the 0.82 scaling factor significantly improved agreement between GK flux and HIPPO observations, based on simulations using one ATM (a different MIROC4-ACTM configuration). However, our results show that applying this 0.82 scaling factor actually leads to an underestimation of modelled column-mean APO SCA when comparing with the combined HIPPO, ORCAS, and ATom observations at high latitudes in both hemispheres.

15. Page 44, line 1025: "Naegler et al. (2006)" should read "Naegler et al. (2007)".

It should refer to the following manuscript. We corrected the error.

Naegler, T., Ciais, P., Rodgers, K., and Levin, I.: Excess radiocarbon constraints on air-sea gas exchange and the uptake of CO2 by the oceans, Geophys. Res. Lett., 33, https://doi.org/10.1029/2005GL025408, 2006.

16. Page 60, line 1410-1413: "Kenneth et al., 2021" is not cited in the manuscript.

We intended to refer to Schuldt et al., 2021, which is now included in the 'References' section.

Reviewer #2

This manuscript presents results from a forward model intercomparison project using Atmospheric Potential Oxygen (APO), comparing eight models/model variants and three APO flux products. The focus of the study is on evaluating uncertainty in atmospheric transport models (ATMs) by comparing to surface, shipboard and aircraft observations with a focus on latitudinal and seasonal variability, annual mean APO, seasonal cycle amplitude and seasonal minimum timing, as well as diabatic mixing rates. Unlike previous studies, which have tended to focus on tracers with land sources/sinks that are more abundant in the northern hemisphere, this study aims to add value to the field by simulating a tracer of surface ocean exchange with significant seasonal atmospheric variability in both hemisphere. Although a previous APO intercomparison study was already conducted in the mid-2000s, there have been substantial advances since in terms of improvements in ATM performance, in ocean flux simulations, and in methods to evaluate diabatic and adiabatic mixing processes, as well as additional observations; hence, the authors were motivated to conduct a more extensive study.

Overall I find this manuscript to be excellently written and executed, with detailed analyses that are well-explained and presented in high-quality figures. I consider this manuscript an excellent fit to the journal and recommend it be published with only minor corrections as follows:

1. Line 25: This definition of APO is not strictly true, but I think if the wording is changed to "APO… can be used as a tracer of air-sea O2 exchange…" or similar then this fixes the issue, which presumably only arises in the abstract for reasons of brevity as the full explanation in the introduction is correct.

We edited the sentence to recognize the contribution of fossil fuel burning to APO.

L25 - 27: Atmospheric Potential Oxygen (APO, defined as $O_2 + 1.1 \times CO_2$) is primarily a tracer of ocean biogeochemistry and fossil fuel burning. APO exhibits strong seasonal variability at mid-to-high latitudes, driven mainly by seasonal air-sea $O_2$ exchange.

2. Line 27: very minor point, but perhaps there should be a "the" before "Atmospheric Potential Oxygen forward Model Intercomparison Project" as per the title?

Edited as suggested

3. Line 52: Strictly speaking, I think the APO equation should have "≈" instead of "=" as this version excludes the oxidation of CH4 and CO molecules as per the original Stephens et al. 1998 publication. This also applies to Equation 1 around line 155.

Edited as suggested.

4. Lines 71-72: Perhaps the citations here should be preceded by "e.g." as this is probably not the complete list?

Edited as suggested.

5. Line 140: This paragraph begins with an explanation of section 3, but perhaps 1-2 sentences of explanation on section 2 would also be helpful?

We edited and added the following sentences.

L143 - 144: In Section 2, we describe APO measurements from surface stations, aircraft, and ships, and the experimental design of APO-MIP1.

L154 - 157: In Section 3.4, we discuss the broader implications of our analysis for developing methods to identify processes that introduce transport biases and for improving atmospheric transport modeling.

6. Line 168: Does the first mention of the unit "per meg" need a citation?

For the per meg unit, we refer to the following manuscript, which is included in the 'References' section.

Keeling, R. F., Manning, A. C., McEvoy, E. M., and Shertz, S. R.: Methods for measuring changes in atmospheric $O_2$ concentration and their application in southern hemisphere air, J. Geophys. Res. Atmospheres, 103, 3381–3397, https://doi.org/10.1029/97JD02537, 1998.

7. Line 339: Schuldt et al., 2021 is missing from the reference list.

The following reference was added.
Schuldt, K. N., Mund, J., Luijkx, I. T., Aalto, T., Abshire, J. B., Aikin, K., Andrews, A., Aoki, S., Apadula, F., Baier, B., Bakwin, P., Bartyzel, J., Bentz, G., Bergamaschi, P., Beyersdorf, A., Biermann, T., Biraud, S. C., Boenisch, H., Bowling, D., Brailsford, G., Chen, G., Chen, H., Chmura, L., Clark, S., Climadat, S., Colomb, A., Commane, R., Conil, S., Cox, A., Cristofanelli, P., Cuevas, E., Curcoll, R., Daube, B., Davis, K., De Mazière, M., De Wekker, S., Della Coletta, J., Delmotte, M., DiGangi, J. P., Dlugokencky, E., Elkins, J. W., Emmenegger, L., Fang, S.,

Fischer, M. L., Forster, G., Frumau, A., Galkowski, M., Gatti, L. V., Gehrlein, T., Gerbig, C., Gheusi, F., Gloor, E., Gomez-Trueba, V., Goto, D., Griffis, T., Hammer, S., Hanson, C., Haszpra, L., Hatakka, J., Heimann, M., Heliasz, M., Hensen, A., Hermanssen, O., Hintsa, E., Holst, J., Ivakhov, V., Jaffe, D., Joubert, W., Karion, A., Kawa, S. R., Kazan, V., Keeling, R., Keronen, P., Kolari, P., Kominkova, K., Kort, E., Kozlova, E., Krummel, P., Kubistin, D., Labuschagne, C., Lam, D. H., Langenfelds, R., Laurent, O., Laurila, T., Lauvaux, T., Lavric, J., Law, B., Lee, O. S., Lee, J., Lehner, I., Leppert, R., Leuenberger, M., Levin, I., Levula, J., Lin, J., Lindauer, M., Loh, Z., Lopez, M., Machida, T., et al.: Multi-laboratory compilation of atmospheric carbon dioxide data for the period 1957-2020; obspack_co2_1_GLOBALVIEWplus_v7.0_2021-08-18, , https://doi.org/10.25925/20210801, 2021.

8. Line 382: I think it is the difference between the models and the multi-station global mean shown by the colours in figure 4, not the model errors?

We replaced 'errors' with 'model-observation differences'.

9. Line 383: Meridional gradients in the observations are clearer in Figure 5. Also, perhaps for clarity the figure 4 caption should mention that the sites are organised by latitude from south (left) to north (right)?

We intended to point to Figure 5 for the meridional gradient. We also edited the figure 4 caption to mention the organization of sites.

10. Line 384: I think it would help the reader to identify more clearly where this southern tropical bulge is being observed (presumably it is in the values for SMO?)

We edited this sentence as follows.
L390 - 393: Observations show clear meridional gradients in APO annual means (Fig. 5A), with higher values in the Southern Hemisphere than Northern Hemisphere, and a southern tropical "bulge" evident at SMO and in the airborne data centered on 15°S.

11. Lines 482-483: I disagree with this description of the figure, as it seems to me that Jena is over-estimating SCA in the northern mid-latitudes, while CESM agrees quite well and DISS is under-estimating at all latitudes. Perhaps instead the authors are referring to the annual mean panels, where CESM and DISS seem to over-estimate at those latitudes?

We intended to refer to annual mean values. We corrected this sentence below.

L494 - 495: Simulations using CESM and DISS flux products show larger annual mean values in the northern mid-latitudes (40 - 60°N).

12. Figure 6: I think the plot labels here should read (A), (B), (C) as per the caption? Also it might be better to place the legend at the bottom of the plot rather than below the (A) panel label.

We corrected the error and moved this figure legend as suggested.

13. Line 1188: Should be "References"

Fixed.

14. Line 1410: The Kenneth et al 2021 reference is not mentioned in the text anywhere.

We intended to refer to Schuldt et al. (2021), which is now included in the references.

15. Lines 1448-1453 and 1495-1503: There are two different Long et al (2021) references and two different Nevison et al. (2008) references that both need "a" and "b" suffixes.

For two Nevison et al. (2008), we want to refer to the following paper at both places. The error was fixed.

Nevison, C. D., Mahowald, N. M., Doney, S. C., Lima, I. D., and Cassar, N.: Impact of variable air-sea $O_2$ and $CO_2$ fluxes on atmospheric potential oxygen (APO) and land-ocean carbon sink partitioning, Biogeosciences, 5, 875–889, https://doi.org/10.5194/bg-5-875-2008, 2008.

For two Long et al. (2021), we have edited them as 2021a and 2021b.

16. General comments on the figures: Overall these are excellent, however, I'm not sure the colour schemes used are fully accessible? So the authors might want to consider alternatives, and perhaps also the use of additional symbols/line types. Resolution could also perhaps be improved for Figures 1, 6, and 8 in particular. The axes labels in Figures 6, 8 and 10 are also quite small and it might be beneficial to make these larger and easier to read. Finally, is it possible to either

add error bars to the observations in Figures 3, 5, and 6, or if these are not easily visible in the plots, is it possible to give an indication of the typical observational error in the figure captions. This would help to place some of the model differences into context when these are small, especially for readers who are not overly familiar with APO and its typical measurement errors.

We have increased the font sizes of axis labels and titles in Figures 6, 8, and 10, and corrected errors in the panel labels of Figure 6. The color scheme employed is developed by Paul Tol and designed to be accessible to colorblind individuals (see https://davidmathlogic.com/colorblind/#%23332288-%23117733-%2344AA99-%2388CCEE-%23DDCC77-%23CC6677-%23AA4499-%23882255). High-resolution figures in PDF format will be provided with the final manuscript submission.

Our model-observation comparisons are based on subsampling model data at the exact times and locations of observational data; therefore, observational uncertainty stems only from measurement imprecision. However, the measurements are highly precise. For airborne data, there is a ±2.1 per meg (1-σ) imprecision for each 10-second measurement. Due to the sufficient number of 10-second data points, this imprecision is averaged out when calculating column averages and/or 10-degree latitude band averages. Since only one instrument measures $O_2/N_2$ at surface stations and along aircraft flight tracks, we cannot access the systematic errors in our measurements. The overall uncertainty in observed APO annual means and seasonal cycles at each station, and in airborne column averages for each 10 degree latitude band, is negligible compared to the model spread. We have added the following sentence to acknowledge the small measurement uncertainty.

L1002 - 1005: The resolved APO annual mean and seasonal cycles have negligible measurement uncertainty compared to model spread because we average data over long time series for stations and over large spatial domains for aircraft and ships, effectively reducing the already small short-term instrument imprecision.